# *bit2bit*: 1-bit quanta video reconstruction by self-supervised photon location prediction

Yehe Liu[1,2]    Alexander Krull[3,*]    Hector Basevi[3]    Aleš Leonardis[3]    Michael Jenkins[1,2,*]

[1]Case Western Reserve University    [2]OpsiClear LLC    [3]University of Birmingham

{yehe, mwj5}@case.edu    {a.f.f.krull, h.r.a.basevi, a.leonardis}@bham.ac.uk    Joint supervision[*]

## Abstract

Quanta image sensors, such as single-photon avalanche diode (SPAD) arrays, are an emerging sensor technology, producing 1-bit arrays representing photon detection events over exposures as short as a few nanoseconds. In practice, raw data are post-processed using heavy spatiotemporal binning to create more useful and interpretable images at the cost of degrading spatiotemporal resolution. In this work, we propose *bit2bit*, a new method for reconstructing high-quality image stacks at the original spatiotemporal resolution from sparse binary quanta image data. Inspired by recent work on Poisson denoising, we developed an algorithm that creates a dense image sequence from sparse binary photon data by predicting the photon arrival location probability distribution. However, due to the binary nature of the data, we show that the assumption of a Poisson distribution is inadequate. Instead, we model the process with a Bernoulli lattice process from the truncated Poisson. This leads to the proposal of a novel self-supervised solution based on a masked loss function. We evaluate our method using both simulated and real data. On simulated data from a conventional video, we achieve 34.35 mean PSNR with extremely photon-sparse binary input (<0.06 photons per pixel per frame). We also present a novel dataset containing a wide range of real SPAD high-speed videos under various challenging imaging conditions. The scenes cover strong/weak ambient light, strong motion, ultra-fast events, etc., which will be made available to the community, on which we demonstrate the promise of our approach. Both reconstruction quality and throughput substantially surpass the state-of-the-art methods (e.g., Quanta Burst Photography (QBP)). Our approach significantly enhances the visualization and usability of the data, enabling the application of existing analysis techniques.

## 1   Introduction

Quanta image sensor (QIS) (e.g., SPAD arrays [1, 2]) offer unique advantages over other image sensors for capturing fast temporal dynamics in low-light settings, as they can detect individual photons within nanoseconds exposures [3]. This allows for the measurement of ultra-fast phenomena, such as fluorescence lifetime [4], time-of-flight [5], and other time-resolved processes [6]. Quanta image data is different from traditional digital images. The QIS raw data often consists of sparse detection events described in 1-bit arrays (Fig. 1). In each frame, pixels only indicate the presence or absence of a photon without intensity information [7]. This makes the data non-interpretable or usable as traditional images. A common approach to make effective use of the binary data is to bin the image in the time domain[8]. While this can provide clear images (Fig. 1c), it comes at the cost of losing the valuable high-temporal-resolution information offered by QIS. In this paper, we present a novel self-supervised approach that directly denoises binary photon detection events from a QIS, reconstructing a clean video at the original temporal resolution.

38th Conference on Neural Information Processing Systems (NeurIPS 2024).

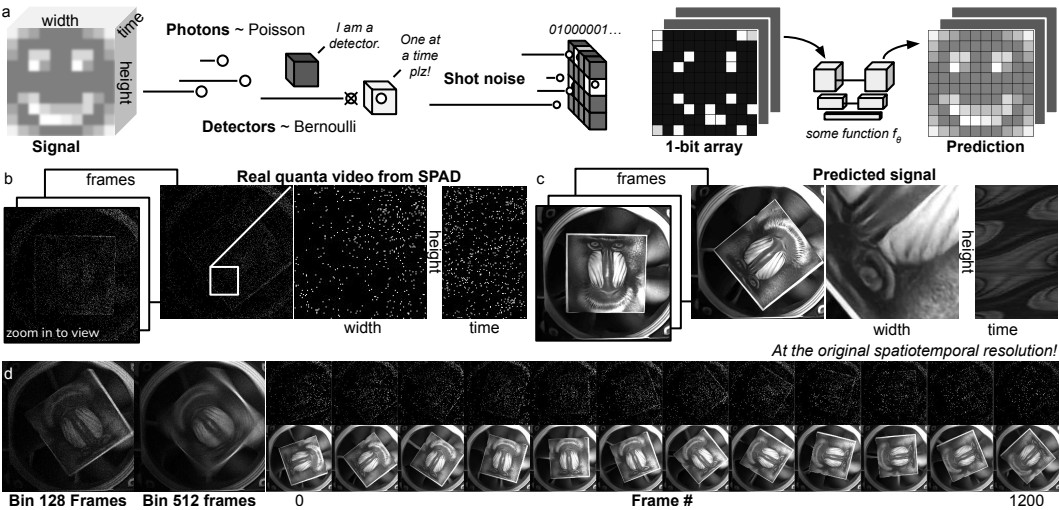

Figure 1: **Visualization of the reconstruction task.** a. A signal in spacetime generates discrete photons through a Poisson process. Real detectors can only count one photon at a time. The discrete nature of photons and the discrete counting process introduce shot noise, resulting in a sparse binary map. Our goal is to predict the underlying signal from this information-sparse data. b. Real SPAD raw data captured by a detector. The highlighted box indicates a zoomed-in region, revealing sparse photon detection events. To the right is a cross-section of the time-height dimensions, showing a similar binary noisy pattern. c. Our method produces the video from the data in b at the original spatiotemporal resolution (Video S1). d. Left: effect of accumulating raw data frames directly, showing shot noise and motion artifacts. Right: additional keyframe pairs are provided for reference.

Generative Accumulation of Photons (GAP) recently introduced an effective self-supervised method that denoises shot noise corrupted photon counting images by splitting images into paired data training pairs based on Poisson statistics [9]. The self-supervised method is valuable to scientific imaging applications, where very often ground truth data is unavailable and cannot be simulated. However, this method is ineffective for binary images, where it creates significant artifacts due to the non-Poissonian nature of photon statistics in 1-bit data [10, 11]. This inspired us to develop *bit2bit*, a new method that specialized for binary data. Our key new insight is a masking strategy that hides the complementary dependency within the training pairs, which effectively addresses the problem. We further extend our method to 3D to leverage the information in both space and time, substantially improving the reconstruction quality. In addition to these measures, we also explored different architectures and sampling parameter spaces to reduce the significant issue of overfitting.

**Problem statement and claims.**    The goal of this work is to predict the underlying spatiotemporal signal from a stack of quanta raw data. Each frame in the stack is a 1-bit 2D array representing locations where at least one photon is detected during the exposure period (Fig. 1a). Our method is based on the following 2 assumptions: 1) Each detection event is independent and follows truncated Poisson statistics. 2) The underlying signal exhibits some local and global spatiotemporal structures that can be described by a model.

Given this challenging reconstruction problem, we claim the following main contributions:

1. Developing a self-supervised method that denoises photon-sparse binary quanta images, which effectively handles the binary nature of the data through a novel masking strategy.
2. Enhancing the performance of reconstruction by leveraging the temporal information that is readily available in most quanta image data.
3. Providing insights on stochastic sampling strategies, network designs, and regularization techniques to manage overfitting and improve the reconstruction quality.
4. Presenting a novel dataset with real and simulated 1-bit SPAD data to support further research and quantitative evaluation of quanta image processing.

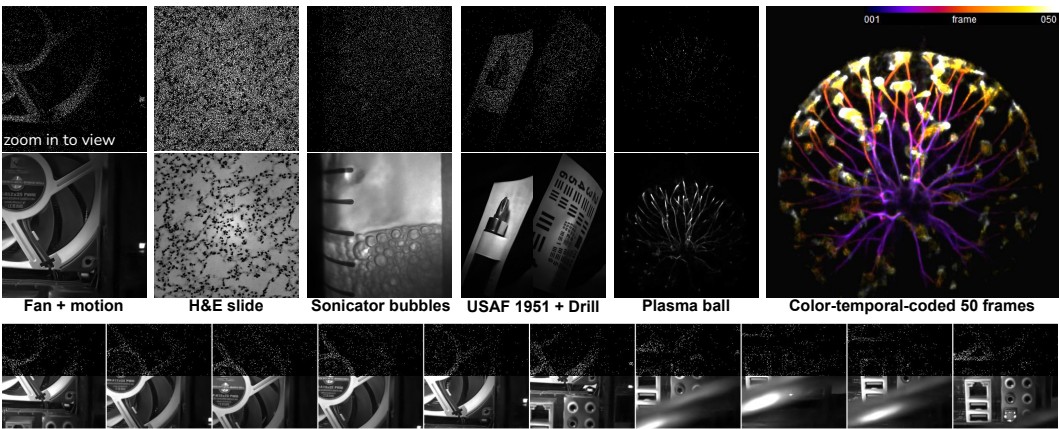

Figure 2: **Example Results from Our Method Using Real SPAD Data** The top row displays raw SPAD data. The middle row shows the corresponding reconstructions using our method. **CPU Fan + motion:** Imaged under camera motion. Additional paired raw data and reconstruction keyframes are shown below. **H&E slide:** Moving under a microscope. **Sonicating bubbles:** Humidifier generates bubbles, water droplets, and mist. **USAF 1951 + drill:** Resolution target spinning on a drill. **Plasma ball:** Firing plasma. A color-coded accumulation of 50 frames is shown on the right. [More in supp]

## 2   Related Work

**Quanta image reconstruction.**   A common quanta image reconstruction task is finding the optimal transformations of individual 1-bit frames to minimize motion blur prior to applying binning [8, 12–14]. Noise is another problem being addressed, especially in SPAD-based 3D reconstruction [15–17]. Some higher-level tasks address the motion deblur of the binned data using specially engineered deep learning methods because the mixing of shot noise renders traditional debluring methods ineffective [18]. Besides direct image reconstruction, there are also efforts to perform traditional computer vision (CV) tasks directly to quanta image data (e.g., image classification) [19–21]. The task described in this work - reconstructing single binary frames at original temporal resolution and addressing both shot noise and motion blur - is challenging and has not been well explored. Only one previous work explored a similar task using the supervised method with ground truth data [22], in contrast to our self-supervised method.

**Self-supervised denoising.**   Convolution neural networks (CNN) can be trained to denoise images by minimizing the difference (e.g., mean squared error (MSE)) between the prediction and a reference target [23, 24]. This is usually referred to as the minimal mean squared error (MMSE) denoising. Noise2Noise (N2N) demonstrated that it is possible to minimize the MSE loss between a pair of noisy images and still get quality denoising results without knowing the noise-free prior [25]. However, N2N still requires real data pairs with the same underlying signal, which are often not obtainable. More recently, self-supervised denoising methods have been introduced to perform denoising from a single noisy image. The methods create input and target training pairs from the original noisy image by assuming the augmentations minimally affect the underlying signal. Noise2Void (N2V)(2) predicts the value of random pixels in a noisy image by hiding these pixels during training and only using the surrounding context to minimize loss only from the hidden pixels [26, 27]. Noise2Self (N2S) presented a similar concept that uses a moving grid mask [28]. Noisier2Noise adds more noise to the input to alter the original noise [29]. Neighbour2Neighbour, Noise2Fast, etc., assume pixel-wise independence of noise and create image pairs from neighboring pixels [30, 31].

However, the binary, sparse nature of quanta image creates a special, challenging case. A recent self-supervised denoising method, GAP [9], effectively addressed shot noise corruption in photon counting images by Poisson-statistic-based resampling. GAP introduced the idea of splitting the data by pixel-wise binomial sampling to create nearly unlimited training pairs, which inspired our work. However, implementing GAP directly on binary quanta images resulted in extremely poor

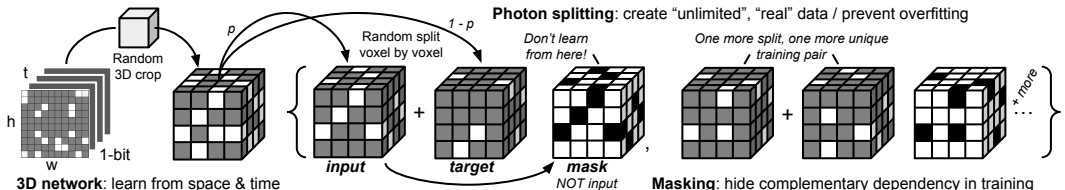

Figure 3: **Overview of the sampling/masking strategy.** The raw data is processed in 3D to use space and time similarly. Data pairs are created by random 3D crop from the raw data, then randomly split the positive values into an input or a target matrix. The split ratio is controlled by a parameter p. A mask is created by flipping the bits in the input image, which prevents gradient back-propagation from locations of 1s in the input. This process is repeated indefinitely, each time creating a new pair of data equivalent to independent observations from the underlying signal.

reconstructions. This motivated us to identify the root cause of the problem and develop a specialized self-supervised method for sparse binary image reconstruction.

## 3 Theories and methods

**Brief review of GAP.** Here, we summarize the self-supervised denoising approach described in GAP [9]. GAP is based on the assumption that observed image $\mathbf{x}$ is a shot noise corrupted version of the clean image $\mathbf{s}$, with each pixel $x_i$ drawn from a Poisson distribution parameterized by the clean signal at the pixel $s_i$, where $i$ indicates the pixel index. GAP proposes to split the noisy image $\mathbf{x}$ by sampling from a binomial distribution randomly assigning photons to an input image $\mathbf{x}_{\text{inp}}^k$ and a target image $\mathbf{x}_{\text{tar}}^k$. The images are conditionally independent given the underlying signal $\mathbf{s}$, so can be treated as two independent observations, which can then be utilized in a N2N-style training procedure [25] to train a neural network $f(\mathbf{x}; \theta)$ with parameters $\theta$. By attempting to predict $\mathbf{x}_{\text{tar}}$ from $\mathbf{x}_{\text{inp}}$, the procedure trains a network to find the MMSE estimate for the clean signal. It uses a cross-entropy loss over pixels

$$L(f(\mathbf{x}_{\text{inp}}^t; \theta), \mathbf{x}_{\text{tar}}^t) = \sum_i^n \mathbf{x}_{\text{tar}}^t \ln \frac{\exp(f(\mathbf{x}_{\text{inp}}^t; \theta)_i)}{\sum_j^n \exp(f(\mathbf{x}_{\text{inp}}^t; \theta)_j)}, \tag{1}$$

viewing the problem as a classification task, trying to predict the photon locations in the target image. Here, $f(\mathbf{x}_{\text{inp}}^t; \theta)_i$ corresponds to the logit output for pixel $i$. The authors show that in the limit, their loss is equivalent to the MSE loss, which is more frequently used in N2N training.

**Photon counting.** While the assumption of the Poisson shot noise model in GAP seems reasonable for photon counting, it often does not accurately hold for truncated QIS data such as SPAD. We can understand this by modeling photon counting as a Poisson point process and considering each counted photon as an element of a set defining this process. Subsequently, we can model the compound processes and the sub-processes within the framework of point process statistics and differentiate between photon counting and sector activation events (e.g., binary activations in each frame). This broader insight is discussed in the Appendix. This section focuses on the practical implementation of this insight in self-supervised denoising.

**Quanta image generation.** Instead of truly counting photons, SPAD (and other quanta imaging methods) acquire a series of binary images $\mathbf{x}^t$, with each pixel $x_i^t \in \{0, 1\}$ indicating whether photons have been detected or not. Unfortunately, this means that the number of photons hitting the pixel is not recorded in a unit detection window, and we cannot know if an active pixel $x_i^t = 1$ receives single or multiple photons during one exposure. Formally, pixel values $x_i^t$ are drawn from a Bernoulli distribution with parameter $\rho = 1 - e^{-s_i}$, with $e^{-s_i}$ corresponding to the probability of zero photons hitting a pixel according to the underlying Poisson distribution (Appendix A.1). For small $s_i$, individual pixels rarely receive multiple photons during an exposure. Thus, it is convenient to consider $\rho_i = 1 - e^{-s_i} \approx s_i$ [11].

In order to generate images that are visually pleasing and easier to analyze, the sparse binary data is usually processed by summing multiple frames $\mathbf{x}^t$, effectively counting the number of detection

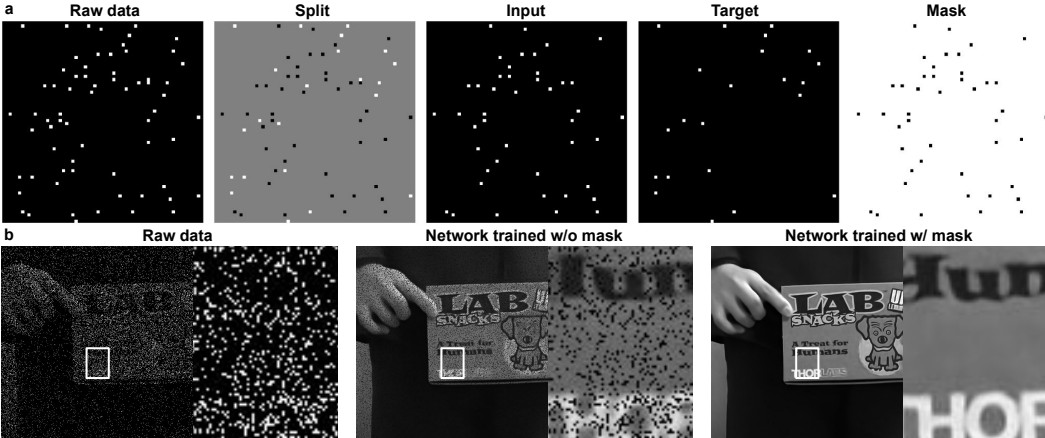

Figure 4: **Real data examples of photon splitting and the effect of the masked loss** a. Example of splitting a randomly selected quanta image raw data frame. The **Raw data** consists of only binary pixels indicating the location of the photon counting event. The **Split** indicates the **Input** (black) and **Target** (white) of the split. The **Mask** is calculated by inverting the **Input** and is applied to the loss. b. Comparison of the training results with unmasked and masked loss. Without the masked loss, the network learns that whenever a pixel location has a photon in the input, it never has a photon in the target. The deterministic relationship leads to the artifacts. The pixel locations where the input is 1 appear dark in the network output. The masked loss effectively addresses the problem.

events for each pixel [7]. Unfortunately, this aggregation over time leads to an inevitable loss of temporal resolution, as information about the time in which a photon hit is lost. It is important to note that the noise in such data is not Poisson distributed but instead follows a binomial distribution. Unfortunately, this means GAP, which relies on this assumption, cannot be directly applied to SPAD data. In the Appendix (Photon detection), we give an alternative theoretical perspective on this phenomenon by modeling photon counting as a Poisson point process with sub-processes to understand the difference between photon counting and the counting of photon detection events. While it is true that we can produce approximate Poisson statistics by summing a large number of frames, this comes at the cost of significantly reduced time resolution.

**Splitting of 1-bit quanta data.**   This work explores an alternative approach, attempting to directly split the recorded binary images without temporal aggregation. We split the recorded binary image according to GAP [9], randomly assigning each photon detection event in $\mathbf{x}^t$ to $\mathbf{x}^t_{\text{inp}}$ or $\mathbf{x}^t_{\text{tar}}$ with a probability $p$ (Fig. 3). Assuming low light intensity $s_i$ this means that the probability for finding a photon detection event at a pixel is $\rho_{i,\text{inp}} \approx ps_i$ and $\rho_{i,\text{tar}} \approx (1-p)s_i$. In order to enable the use of temporal information, we apply the same not to 2dimensional images but to 3D space-time-volumes $X = (\mathbf{x}^1, \ldots, \mathbf{x}^T)$, splitting them into input $X_{\text{inp}} = (\mathbf{x}^1_{\text{inp}}, \ldots, \mathbf{x}^T_{\text{inp}})$ and target volumes $X_{\text{tar}} = (\mathbf{x}^1_{\text{tar}}, \ldots, \mathbf{x}^T_{\text{tar}})$.

However, unlike for true Poisson distributed photon counts, $\mathbf{x}^t_{\text{inp}}$ and $\mathbf{x}^t_{\text{tar}}$ cannot be considered conditionally independent. Since a photon detection event in $x^t_i = 1$ can only be assigned to either $\mathbf{x}^t$ to $\mathbf{x}^t_{\text{inp}}$ or $\mathbf{x}^t_{\text{tar}}$, an event $\mathbf{x}^t_{i,\text{inp}} = 1$ in the input image must necessarily mean that corresponding pixel in the target image does not contain an event $\mathbf{x}^t_{i,\text{tar}} = 0$. As a consequence, naively using $\mathbf{x}^t_{\text{inp}}$ and $\mathbf{x}^t_{\text{tar}}$ as training pairs for a denoiser, will lead to the artifacts, with the network predicting dark values for any pixel which has a photon detection event in its input.

**Masked loss function photon prediction for quanta images.**   To remedy this effect of correlated input and target images, we propose an adapted loss function. Considering that our data is sparse, that is, only very few pixels contain photon detection events, we avoid the problem by excluding

pixels that contain a photon detection event in the input image using the loss

$$L\big(f(X_{\text{inp}};\theta), X_{\text{tar}}\big) = \sum_{i,t}^{n,T}(1 - x_{i,\text{inp}}^t)x_{i,\text{tar}}^t \ln \frac{\exp(f(X_{\text{inp}};\theta)_i^t)}{\sum_{j,\tau}^{n,T}\exp(f(X_{\text{inp}};\theta)_j^\tau)}, \tag{2}$$

with $f(X_{\text{inp}};\theta)$ referring to the logit output of the denoiser network and $f(X_{\text{inp}};\theta)_i^t$ referring to the value at the pixel/time index $i, t$ of the output. Note that, while the masking is achieved by multiplying $(1 - x_{i,\text{inp}}^t)$ the remainder of the function corresponds exactly to the cross entropy loss over pixels from Eq. 1.

**Implementation details.** We view each binary quanta image dataset as a 3D array with two spatial dimensions and a temporal dimension. We randomly crop each sample $\mathbf{x}$ from the volume (Fig. 2). $\mathbf{x}_{\text{inp}}$ is sampled from a volume $\mathbf{x}$ by Bernoulli sampling with dropout probability $p$ defined based on desired strategy. We then calculate the target $\mathbf{x}_{\text{tar}}$ by subtracting the input $\mathbf{x}_{\text{inp}}$ from $\mathbf{x}$. The mask $\mathbf{x}_{\text{mask}}$ is computed from the input by $\mathbf{x}_{\text{mask}} = 1 - \max\{\mathbf{x}_{\text{inp}}, 1\}$ at runtime. In the binary case, the mask is equivalent to the bitwise complement of the input $\neg\mathbf{x}_{\text{inp}}$ (Fig. 3). We multiply the mask by the output of the network and the target before the cross entropy loss is calculated, which is equivalent to the loss function discussed in the previous section.

**Selecting photon splitting variable p.** The Bernoulli sampling probability $p$ is an important new hyperparameter introduced in the sampling process. Its value and selection strategy can drastically impact the performance of production. We will demonstrate this in our experimental section. If we only work at a fixed input signal level (e.g., mean photon count), we can use a fixed $p$ to generate training pairs. However, it is necessary to reduce the photon count to the same signal level for inference. A fixed large or small $p$ can lead to over-fitting and slow training. In extreme cases, the model will recreate the target if $p = 1$, and the target is always empty if $p = 0$. A single input at the reduced signal level does not contain all the available information. We could resample multiple copies of the input and combine the inference results, but this strategy requires a longer inference time. Alternatively, we can randomly pick $p$ from a desired range, which allows the model to adapt to different signal levels and produce acceptable results from all the inputs within the range. If we use a large $p$ close to 1 as the upper limit (e.g., $1 - 10^{-6}$), we should be able to use raw data directly as the input, which is more efficient in practice.

## 4 Experiments

**Network architecture.** We used a 3D ResUNet [32] with an option to switch the network to 2D at a specific depth. The primary reason for this design choice is to handle the rapid increase of the receptive field as the depth increases, which eventually becomes bigger than the sampled area. The network will learn a lot from the padded areas during training. In the photon sparse binary case, this problem is more prominent, as the 0 padding is not different from the real data. To avoid significant weight from the padded area, a large crop window size is necessary to achieve better results. However, this can easily lead to memory issues. We can prevent the receptive field from growing in the temporal dimension by switching from 3D to 2D at the desired depth. This allows us to use a smaller window size in the temporal dimension and a larger window size in the spatial dimension. A detailed diagram of the network architecture is shown in Fig. S1. For most experiments, we set the network depth at 5, with only the first 2 levels being 3D to control the size of the receptive field. We used 32 initial input features, which provide a balance between performance and memory efficiency. The number of features doubles at each depth. Group normalization of 8 is applied at each convolution [33]. We use GeLU for the activation function and pixel shuffling for up-sampling [34, 35].

**Training.** Models were trained using the ADAMW optimizer for 150 epochs, with 250 steps per epoch and 4 batches of random crops of 32x256x256 (TXY) per step [36]. The large patch size is desired to prevent performance degradation due to substantial padding weights. However, 3D UNet with a large patch size is not memory-efficient, and data transfer is a bottleneck. Sometimes, only 2 batches of the crop size can be trained at a time while leaving a substantial memory unused with 24GB VRAM. Therefore, we implemented gradient check-pointing [37], which allowed us to double the batch size and utilize the majority of the VRAM. We primarily used Nvidia 3090/4090 for training, and each training takes about 6-10 hours until the loss curve stabilizes.

**Inference.** We run patch-based inference on a GPU using the original image width and height, maximizing the frame count for each inference. With 24GB VRAM, we can use a 48x512x512 volume for each inference with the typical 5-depth ResUNet mentioned above. The inference speed is above 3 volumes per second (150 fps) on a NVIDIA RTX 4090 GPU. If the training uses $p_{\max} \approx 1$, the raw data is used directly as input. Otherwise, we randomly reduce the photon count in the data by a factor of $p_{\max}$ through photon splitting. For a 10-shot inference, we Bernoulli sample 10 times from the raw input at a fixed $p$ value, run inference independently using the model trained at the $p$ value, and calculate the mean of all the outputs.

**Data.** To evaluate our method quantitatively and qualitatively, we use a synthetic video with simulated noise consisting of 3990 frames and a total of 7 real SPAD videos, each containing 100k-130k frames. Additionally, we use a real video with 100k frames published in [8]. Simulated data is essential for quantitative assessment as QIS ground truth is impractical to get. Assuming reference images/videos are ground truth with no shot noise and the pixel values $n_i$ are proportional to the Poisson rate $\lambda_i$, we calculate a variable $q$ based on the selected mean counting rate $\overline{\lambda} = q(\frac{1}{M} \sum_{i=1}^{M} n_i)$, where $\frac{1}{M} \sum_{i=1}^{M} n_i$ is the mean of the reference. We then run pixel-wise Poisson sampling with $\lambda_i = n_i p$ and clip between 0 and 1 to simulate the 1-bit quanta data. A ground truth reference video of a person shaking a ThorLab's box was acquired from iPhone 15 slow motion mode at 240 fps (Fig. S2). The simulation was using $\overline{\lambda} = 0.0625$. After the clipping, the resulting simulation has 0.0590 photons per pixel. Real QIS data were acquired using a SPAD512S camera (Pi Imaging Technology, Switzerland)[2]. Each dataset has 100k-130k binary frames. We created scenes to represent a combination of different imaging conditions, including low-signal, high-signal, high-contrast, high-ambient-light, moving camera, moving object, linear movement, random movement, combined movement, ultra-fast events, and stochastic events (Fig.S6, S7, S8, S9, S10, S11, S12). We made the simulated data, real SPAD data, and reference results available to the community*.

**Comparisons of different methods.** We tested supervised training, N2N, N2V, and the original GAP with simulated data. To ensure fair comparisons, we incorporated the same network architectures, hyperparameters, and training steps into individual baselines. All implementations are in 3D except for GAP. Inference was conducted on the first 512 frames, and the data was normalized by the mean. PSNR and SSIM were calculated frame-wise, and the statistics were derived from the entire output stack. We tested N2N in three different ways: Using two independent simulated samples for training. Sampling two independent image stacks using the proposed splitting method (with $p = 0.5$, just once before training). Using the same samples but applying the masking strategy during training. We tested N2V using the original masking strategy. The N2V masks were created from 1000 random points in space. A median filter was applied to these points in the input. The minimum distance between the points was smaller than the median filter size, and loss was only computed from these points. We tested the original GAP in 2D with and without masking. Finally, we evaluated our method with and without masking.

## 5 Results and discussions

**Findings from the comparisons.** Tab. 1 summarizes key numerical and visual results. The supervised methods perform the best, which reflects the approximate upper limit of the network's performance. In reality, the ground truth is often not available, and high SNR data is only available from binning, which degrades time resolution and blends dynamics. N2N with 2 independent samples created a granular pattern on the images, as well as increasing validation loss since the beginning of the training (Fig. S16). Besides, two independent quanta data with the same underlying signal do not exist. One way the address this with N2N is to use adjacent frames, but this reduces the temporal resolution and introduces motion artifacts (e.g., Fig. S3: Interframe N2N). Without the mask, the image shows a strong pepper noise. After implementing the mask, it generates a smooth image, but the granular pattern persists. We also tested our method, which is similar to GAP in 3D. Masking addresses the image artifact problem. The 3D implementation led to substantially improved PSNR and image quality. N2V also produced poor reconstruction results. The result from the original 2D GAP produced also has the presence of pepper noise, similar to our implementation with large fixed p values. Implementing the mask in the GAP resolved the problem. The images are smooth but lack

---

*https://drive.google.com/drive/folders/1M5bsmsaLBkYmO7nMUjK5_m71RonOp-P9

Table 1: Results from comparison experiments.

| Method | Mask | 2D\3D | PSNR | SSIM | Denoised patch |
|---|---|---|---|---|---|
| Raw | n/a | n/a | 4.81/0.55 | 0.017/0.006 |  |
| Supervised | N | 3D | 36.51/1.56 | 0.976/0.006 |  |
| N2N (2-sample) | N | 3D | 32.15/1.16 | 0.931/0.009 |  |
| N2N | N | 3D | 23.14/0.73 | 0.651/0.027 |  |
| N2N | Y | 3D | 28.83/0.58 | 0.843/0.010 |  |
| N2V | N | 3D | 26.16/0.97 | 0.804/0.028 |  |
| GAP | N | 2D | 20.54/0.64 | 0.588/0.059 |  |
| GAP | Y | 2D | 29.09/0.57 | 0.911/0.015 |  |
| bit2bit | N | 3D | 20.89/0.84 | 0.599/0.062 |  |
| bit2bit | Y | 3D | **33.93/0.99** | **0.959/0.007** |  |

fine details. Our method also shows the pepper noise and the fine details are missing when the mask is not applied. Implementing the mask substantially improved the reconstruction quality.

**Inference strategies.** We achieved a PSNR of 33.93 using a network with $p \in [0, 1 - 10^{-6}]$ that takes the original raw data as input and performs inference in one shot. Further, training a network in a less extreme p range $[0.3, 0.7]$ to avoid residual overfitting, then running inferences using 10 different samples sampled from the original data with $p = 7$, and finally combining the data by simple averaging, improved the PSNR to 34.35 (Table. S8). It is worth noting that it should be assumed that the result is specific to this simulated dataset and model configurations.

**1-bit self-resampling.** We noticed that different self-supervised resampling strategies become very similar across different methods in 1-bit. When we create a training pair and largely preserve the original information, we could either keep a positive in one image or move it to another image. Then, the sampling method essentially describes the spatial rule of splitting. For example, GAP uses binomial sampling to randomly drop out some photons. Self2Self [38] uses random binary masks to drop out some pixels, which is equivalent to GAP in 1-bit. N2S and N2V both select some pixels

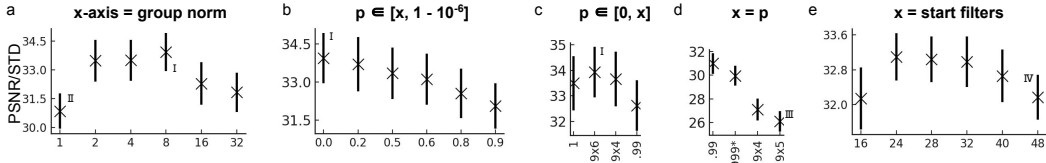

Figure 5: **Results of ablation studies.** a. Group normalization substantially improved the PSNR. b. The choice of lower and c. upper bound of the thinning probability p affects the reconstruction quality. (.9x6:$1 - 10^6$, etc.) d. Fixed large p led to performance degradation despite the proposed single photon prediction suggested in GAP. e. Large model size could negatively impact PSNR. Rome numbers indicate the corresponding images in Fig. S3. Numerical values in Table S2-6.

in the input and use them as a target [28, 27]. The pixels are replaced by a new value using various strategies, such as median, random, and random dropout, which usually remove the selected positive pixel and very occasionally create new positive pixels. If a fixed grid is used, the operation becomes spatial-dependent. In the cases of neighboring pixel-based methods [30, 31], image pairs are created by sub-sampling from a 2x2 grid, equivalent to a balanced spatial-dependent selection followed by downsampling.

**Group normalization.** We noticed that group normalization is *critical* to for this task. Fig. 5a and Table. S2 shows that PSNR increased from 30.9 to 33.5 by using group normalization of 2. In creating the group number to 8 further increased the PSNR to 33.9. We could not implement batch normalization [39], as 24GB VRAM is only sufficient for 1 batch in some use cases (e.g., 512x512x32 window size).

**Over-fitting, stochastic splitting and choices of p.** Overfitting is often unavoidable in self-supervised denoising because the model can learn to fit the noise patterns specific to the finite training data. We noticed granular image artifacts in N2N, N2V, and GAP with large, fixed p-values, which resulted in substantially reduced reconstruction quality (fig. 3). Further analysis indicates that these three examples share a common issue: the input of the model is approximately a fixed binary pattern. In N2N, there is no variation between the input and target data. In N2V, we used a random grid to select the training target, but the grid size was relatively small, and most selected pixels were 0. The input data was very similar to the raw data, with a few photons removed, making the process similar to GAP with a large, fixed p-value.

To analyze the problem, we conducted an experiment using randomly Poisson noise with a small $\lambda$ (Fig. S4). Ideally, the method should produce a uniform plane image with small pixel value variations. Our results show that if a new random input is used for every training step, the output of the network converges to a uniform image as expected, regardless of whether a fixed output is used. If we fix the input and randomize the output, the result is also uniform. However, if we fix both the input and output or if the output contains few photons (e.g., GAP with fixed large p), granular patterns similar to the N2N results start to appear, also reflected by substantially higher standard deviation. Increasing the photon counts in the input reduces shot noise and mitigates the problem. This suggests that diversity in the training pairs is crucial to mitigate overfitting in photon-sparse quanta image data.

**Model size selection.** Increasing model size can negatively impact the performance, potentially due to overfitting. We found a smaller 5-depth network with 24 startup filters outperformed a larger network with 48 filters (Fig. 5e), along with non-significant decaying from 24 to 40 filters. In another comparison, a network with depth 4 and 40 startup filters can achieve comparable performance (PSNR=33.83) as our best 5-depth network with 32 startup filters (33.93) (Table. S7). A 6-depth network (32.55) has a similar performance as a 3-depth network (32.42). The model size parameter space should be explored to balance overfitting and under-representation for different tasks.

**Comparison to QBP.** Quanta burst photography (QBP) is a recognized state-of-the-art quanta image reconstruction method based on spatial-temporal hierarchical alignment and frequency domain combination [8]. We selected a challenging non-static scene from the original work for comparison, where a person is playing the guitar. The results are shown in Fig. 6 and Video S2. It is important to

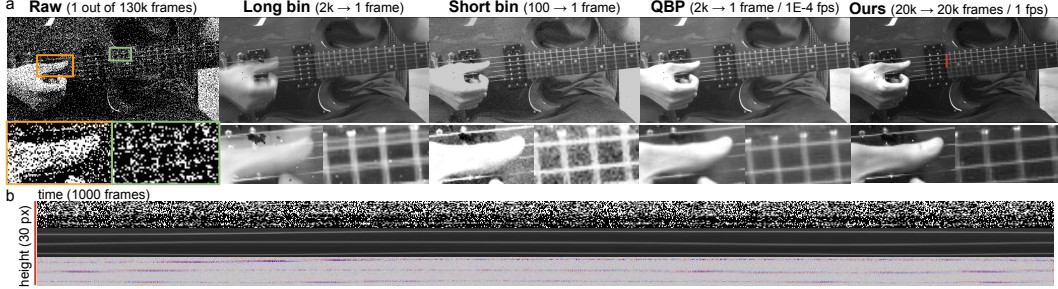

Figure 6: **Comparison of our method to QBP with real SPAD data presented in the paper.** a) Different rendering of the data. The data is from the original QBP, indicating a dynamic scene with a person playing guitar. Our result is shown on the right. b) Height-time slicing of the raw data and our reconstruction. Top: raw data. Middle: our reconstruction, showing the top 3 strings. Bottom: the difference between adjacent frames, indicating sub-pixel movements of the string.

note that QBP reconstructs 1 image from 100-10,000 frames in relatively static scenes, each taking 30 minutes as reported (not optimized) [8]. In contrast, our method can start to produce quality results in less than 1 hour on a 100k-frame dataset. Longer training further improves the results. Our method achieves over 150 fps inference on this dataset, equivalent to processing the entire 100k frames in less than 30 minutes with 50% patch overlap.

## 6 Conclusions

We demonstrated a versatile method for predicting the underlying signal of binary quanta image data, offering a practical solution for quanta image reconstruction where high SNR data and ground truth are unavailable. However, it is necessary to consider the limitations of the method when applying it to real data. Our method assumes a simple Poisson point process model, where the observation is sampled from a noise-free signal. Thus, it is not intended to restore the signal from other types of noise. From a different perspective, the method removes the Poisson noise component from the data, making it easier to identify and correct other noise components. For example, hot pixels on SPAD can be revealed and corrected with median filtering.

A fundamental limitation of self-supervised denoising is the lack of ground truth data, making it difficult to evaluate performance and accuracy in specific cases. In our case, performance could be evaluated by simulating the Poisson process, though these simulations should be based on reasonable assumptions and approximations. Additionally, the method requires offline training for each dataset. While there is potential for real-time reconstruction, the model should be trained on similar scenes.

Our approach can potentially be effective on any binary higher-dimensional spatial data created from Poisson point processes. For example, any data represented by an n-dimensional scatter plot, where points are independent measurements, can be analyzed with this method. Fig. S14 and S15 show the application and benchmark of the method to real and simulated 3D stacks of confocal microscope data with extremely high shot noise, which also demonstrates the performance advantage in 3D.

We also see the task could be generalized for all spatial event point processes as follows [40]. For an n-dimensional signal in spacetime, a finite amount of measurements can be made from disjoint, uniform, and bounded sub-regions. Their collection can be viewed as a Poisson point process. The measurements are clipped between 0 and 1, and the process becomes a Bernoulli process. Given a finite number of such measurements, the task is to predict the distribution of the underlying signal measured by the Poisson rate. We envision that the core concept of creating data pairs by randomly splitting a point process and then hiding their complementary dependencies can potentially be used in other relevant regression models.

Finally, it is worth noting that the method can be misused in research if inappropriate data or statistical assumptions about the noise are made (e.g., the data points are not independent). In such cases, the prediction results can be misinterpreted, leading to incorrect scientific conclusions and potential negative societal impact.

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

# 7 Data and code availability

Simulated and real SPAD data:
`https://drive.google.com/drive/folders/1M5bsmsaLBkYmO7nMUjK5_m71RonOp-P9`

Python raw data loader and cleaner:
`https://github.com/lyehe/spadtools`

SPAD data simulator:
`https://github.com/lyehe/spadsampler`

Source code of the parameterized model with bit2bit:
`https://github.com/lyehe/ssunet`

# 8 Acknowledgment

We thank the following grants for their support of this research: NIH R01EB028635, R01HL126747, S10OD024996, U01EY034693 and R44CA268156.

We thank Axiom Optics for providing the demo unit of the SPAD512s camera.

Special thanks to the DL@MBL (now AI@MBL) program for sponsoring training and collaborations that made this work possible. Cheers Team Restorators and members Anirban Ray, Peter (Hyoungjun) Park, Geneva Schlafly, Guillaume Minet, and Mie Thorborg!

# 9 Competing interests

Yehe Liu and Michael Jenkins are co-founders and owners of OpsiClear LLC.

# A Appendix / Supporting documents

## A.1 Photon counting statistics

**Photon detection.** Here, we are looking at the photon counting from the point process perspective. Imagine an underlying signal indicated by a rate parameter $\lambda$ describing the expected number of photons hitting the detector over a unit of time. Light is quantized, so a hypothetical detector with infinite depth will record $k \sim \text{Poisson}(\lambda)$ photons over each detection window, with $p(k = n) = \lambda^n e^{-\lambda}/n!$, where $n \in \mathbb{N}_0$ is the number of the detected photons. Real detectors can only activate once in each detection window (e.g., $n \in \{0, 1\}$). Therefore, the Poisson point process is reduced to a Bernoulli lattice process with $p(k = 0 \mid \lambda) = e^{-\lambda}$ and $p(k = 1 \mid \lambda) = \sum_{n=1}^{\infty} \frac{\lambda^n e^{-\lambda}}{n!} = 1 - e^{-\lambda}$. With small $\lambda$ (e.g., $< 0.01$), it is convenient to consider $p(k = 0) \approx 1 - \lambda$ and $p(k = 1) \approx \lambda$ in photon-sparse binary data because the chance of missed counts $p(k > 1)$ is negligibly small.

**Photon counting.** From the insight above, it is clear that the "photons" are just "activation events" over some finite detection windows in spacetime. In practice, it is often assumed that $\lambda$ changes minimally across a few windows [11]. The superposition of $n'$ such windows is also a Poisson point process with $\lambda' = n'\lambda$, assuming this homogeneity. If detection is modeled as a more realistic Bernoulli lattice process with $p' = p(k = 1) = 1 - e^{-\lambda}$, summing $n'$ windows is equivalent to binomial sampling $k' \sim \text{Binomial}(n', p')$, which can be approximated by a Poisson distribution with $\lambda' = n'p' = n'(1 - e^{-\lambda})$. Both $\lambda$ and $\lambda'$ describe the underlying signal. It is more trivial to determine $\lambda'$ from multiple windows than $\lambda$ from a single binary image. However, $\lambda_i$ is constantly varying in reality. Through $\lambda' = \sum_i \lambda_i$ or $\sum_i (1 - e^{-\lambda_i})$, we lose the information of individual $\lambda_i$.

**Photon splitting.** Since all points in $N^t$ and $N'$ are independent, we can apply basic point process operation $p$-thinning to randomly remove some points and create new independent sub-processes. The thinning probability $p$ can be selected either manually or randomly to control the size of the new process $N_p$. For $\Lambda$ denoting the original expected count rate, the rate of the thinned process is proportionally reduced to $p\Lambda$. Similarly, the removed points can form another subprocess $N_{1-p}$ with rate $(1 - p)\Lambda$. The new processes $N_p$ and $N_{1-p}$ are independent Poisson processes. Therefore, if we observe a process $N$ over an exposure $\tau$, we can view each split as two real experiments where we observe a first process $N_p$ during $\tau p$ from $p\Lambda$ and, right after, a second process $N_{1-p}$ during $\tau(1 - p)$ from $(1 - p)\Lambda$. GAP originally implemented photon splitting by sampling $X'_p \sim \text{Binomial}(X', p)$ in the binned frames. Similarly, we use a special case $X^t_p \sim \text{Bernoulli}(p \cdot \mathbf{1}(X^t = 1))$ in binary frames.

**Independence of the photon splitting pairs.** $N_p$ and $N_{1-p}$ are both independent Poisson processes. However, they exhibit a complementary dependency due to their definition from the original process, specifically:

$$N_p \cup N_{1-p} = N \quad \& \quad N_p \cap N_{1-p} = \emptyset \tag{3}$$

This implies that for any $x'_{i,p}$, if $x'_{i,p} = n'$, then $x'_{i,1-p} = 0$. For small $\lambda_i$ and large $n'$ (even for $n' = 2$), this is an extremely rare event as $p(x'_i = n') = \prod_{t=1}^{n'} \lambda^t_i$ and $p(x'_{i,p} = n' \mid x'_i = n') = p^{n'}$. Therefore, this was ignored by GAP where the data are accumulations of multiple frames. However, in 1-bit data, all the photons in $N_p$ meet this criteria, as $\forall X_p : X_p = 1 - X_{1-p}$. This also applies to extremely photon-sparse observation when very few photons are detected over a large $n'$ because even with frame bins, we rarely encounter a photon count above 1. We will show this is a crucial problem in binary data, which renders GAP ineffective, and our masking strategy is an effective solution.

**MMSE denoising.** Different underlying signals can lead to an observed value denoted by $N$. Our goal is to predict the signal, described in the compound rate $\Lambda = \{\lambda_i \mid i \in \mathbb{N}^k\}$. We assume a prior distribution $p(\Lambda)$ for the signal and a likelihood function $p(N \mid \Lambda)$ representing the probability of observing $N$ given the signal $\Lambda$. Using Bayes' theorem, we can express the posterior probability and the expected rate of the signals as:

$$p(\Lambda \mid N) = \frac{p(N \mid \Lambda)p(\Lambda)}{\int p(N \mid \Lambda')p(\Lambda') \, d\Lambda'}, \quad \hat{\Lambda} = \int p(\Lambda \mid N)\Lambda \, d\Lambda \tag{4}$$

To estimate the signal from the observation in the sense of MMSE denoising, we look for a function $f_\theta(N)$ and optimal parameter $\theta$ that maps the observation $N$ to the signal $\Lambda$ for each individual dataset. Previous works have shown CNN is effective for this regression task [].

**Signal prediction.** As shown in GAP, we can randomly remove single photons $\phi_i$ from the process $N$ one at a time. The relationship between the removed photons and remaining processes $N_i' = N \setminus \phi_i$ can be used to predict the distribution of upcoming photons. Recall that $\phi_i$ is sampled from a Poisson process with expected rate $\lambda_i \in \Lambda$ and $p(\phi_i \in N \mid \lambda_i) \approx \lambda_i$. We can write:

$$p(\phi_i \in N \mid N_i') = \int p(\phi_i \in N \mid \Lambda, N_i')p(\Lambda \mid N_i') \, d\Lambda = \int p(\Lambda \mid N')\lambda_i \, d\Lambda \tag{5}$$

If we go over all possible locations $i$ and normalize the signal, it becomes clear that predicting the distribution of the next photon is just predicting $\hat{\Lambda}$ in the MMSE sense. We can train model $f_\theta(N)$ to predict this distribution by reducing the cross entropy loss $\mathbf{L}(\theta) = -\sum_i \ln f_\theta(N_i')\phi_i$. Assuming $N_i' \approx N$ in the case of a single photon is negligible in the data, we can apply $f_\theta(N)$ directly.

Training with one photon at a time is time-consuming. And the high similarity between different $N'$ can lead to over-fitting. In practice, we could predict multiple photons at a time using data from photon splitting if we normalize the target:

$$p(N_p \mid N_{1-p}) = \int p(N_p \mid \Lambda, N_{1-p})p(\Lambda \mid N_{1-p}) \, d\Lambda = \int p(\Lambda \mid N_{1-p}) \, p\Lambda \, d\Lambda \tag{6}$$

# B    Appendix / Supplementary tables

Table S1: **The baseline hyperparameters used in training.** This produces the PSNR 33.93/0.99 and SSIM 0.959/0.007 with the simulated data. More details on hyperparameters are included in the supplemental materials in the hp.CSV file.

| | | | | |
|---|---|---|---|---|
| **Model** | **Start features** | **Depth** | **Depth scale** | **3D conv layers** |
| | 32 | 5 | 2 | 2 |
| | **Group norm** | **Up mode** | **Down mode** | **Activation** |
| | 8 | pixelshuffle | maxpool | gelu |
| **Data** | **XY_size** | **Z_size** | **Stack_size** | **Min_p** |
| | 256 | 32 | 1000 | 0 |
| | **Max_p** | **p sampling** | **Flip/Transpose** | **Rotation** |
| | 0.999999 | uniform | FALSE | 0 |
| **Training** | **Batch size** | **Epochs** | **Optimizer** | **Learn Rate** |
| | 4 | 150 | adamw | 0.00032 |
| | **Patience** | **Loss function** | **Masked** | **VRAM** |
| | 20 | Cross entropy | TRUE | 24GB |

Table S2: **Numerical results correspond to Fig. 5a.** The result indicates group normalization is essential for better PSNR and SSIM.

| Group norm size | PSNR | std | min | max | SSIM | std | min | max |
|---|---|---|---|---|---|---|---|---|
| 32 | 31.83 | 1.03 | 29.26 | 35.66 | 0.940 | 0.009 | 0.917 | 0.962 |
| 16 | 32.29 | 1.12 | 28.24 | 36.20 | 0.956 | 0.007 | 0.921 | 0.972 |
| 8 | 33.93 | 0.99 | 29.31 | 36.57 | 0.959 | 0.007 | 0.924 | 0.973 |
| 4 | 33.50 | 1.06 | 28.93 | 36.51 | 0.957 | 0.007 | 0.923 | 0.973 |
| 2 | 33.48 | 1.09 | 28.96 | 36.55 | 0.957 | 0.008 | 0.921 | 0.974 |
| 1 | 30.85 | 0.93 | 27.91 | 33.76 | 0.917 | 0.013 | 0.876 | 0.947 |

Table S3: **Numerical results correspond to Fig. 5b.** Showing the effect of the lower bound of p.

| p range | PSNR | std | min | max | SSIM | std | min | max |
|---|---|---|---|---|---|---|---|---|
| [0.0, 1 - 1E-6] | 33.93 | 0.99 | 29.31 | 36.57 | 0.959 | 0.007 | 0.924 | 0.973 |
| [0.2, 1 - 1E-6] | 33.69 | 1.07 | 29.31 | 36.78 | 0.957 | 0.007 | 0.924 | 0.974 |
| [0.5, 1 - 1E-6] | 33.34 | 1.02 | 29.09 | 36.43 | 0.955 | 0.008 | 0.922 | 0.972 |
| [0.6, 1 - 1E-6] | 33.11 | 1.01 | 28.73 | 36.04 | 0.953 | 0.008 | 0.915 | 0.970 |
| [0.8, 1 - 1E-6] | 32.55 | 0.97 | 28.66 | 35.57 | 0.946 | 0.009 | 0.914 | 0.966 |
| [0.9, 1 - 1E-6] | 32.05 | 0.90 | 28.54 | 34.81 | 0.941 | 0.009 | 0.905 | 0.962 |

Table S4: **Numerical results correspond to Fig. 5c.** Showing the effect of the upper bound of p.

| p range | PSNR | std | min | max | SSIM | std | min | max |
|---|---|---|---|---|---|---|---|---|
| [0-1] | 33.50 | 1.06 | 28.93 | 36.33 | 0.956 | 0.008 | 0.920 | 0.972 |
| [0, 1 - 1E-6] | 33.93 | 0.99 | 29.31 | 36.57 | 0.959 | 0.007 | 0.924 | 0.973 |
| [0, 1 - 1E-4] | 33.66 | 1.07 | 29.00 | 36.48 | 0.958 | 0.008 | 0.920 | 0.974 |
| [0, 1 - 1E-2] | 32.62 | 0.99 | 28.54 | 35.48 | 0.938 | 0.010 | 0.895 | 0.960 |

Table S5: **Numerical results correspond to Fig.5d.** Showing the effect of the fixed p.

| fixed p | PSNR | std | min | max | SSIM | std | min | max |
|---|---|---|---|---|---|---|---|---|
| [1 - 1E-5, 1 - 1E-5] | 26.12 | 0.86 | 25.23 | 29.52 | 0.789 | 0.026 | 0.748 | 0.872 |
| [1 - 1E-4, 1 - 1E-4] | 27.10 | 0.94 | 25.93 | 30.34 | 0.812 | 0.031 | 0.768 | 0.896 |
| [1 - 1E-3, 1 - 1E-4] | 29.95 | 0.84 | 28.39 | 33.19 | 0.907 | 0.013 | 0.865 | 0.943 |
| [1 - 1E-2, 1 - 1E-2] | 30.99 | 0.86 | 28.33 | 34.08 | 0.926 | 0.011 | 0.886 | 0.954 |

Table S6: **Numerical results correspond to Fig. 5e.** Start filter size of 32 produced better PSNR and SSIM in this case.

| Startup filter size | PSNR | std | min | max | SSIM | std | min | max |
|---|---|---|---|---|---|---|---|---|
| 48 | 32.17 | 0.51 | 30.91 | 33.47 | 0.925 | 0.008 | 0.902 | 0.938 |
| 40 | 32.66 | 0.60 | 31.07 | 34.89 | 0.945 | 0.007 | 0.928 | 0.959 |
| 32 | 32.98 | 0.58 | 31.65 | 35.09 | 0.942 | 0.007 | 0.925 | 0.956 |
| 28 | 33.04 | 0.52 | 31.58 | 34.64 | 0.945 | 0.006 | 0.929 | 0.955 |
| 24 | 33.09 | 0.54 | 31.69 | 34.96 | 0.947 | 0.006 | 0.929 | 0.959 |
| 16 | 32.14 | 0.71 | 30.65 | 34.97 | 0.940 | 0.008 | 0.920 | 0.961 |

Table S7: **Results of PSNR and SSIM from different model sizes.** d: Model depth. sf: Start feature size. zc: Number of z(3D) convolutions.

| Model size | PSNR | std | min | max | SSIM | std | min | max |
|---|---|---|---|---|---|---|---|---|
| d=3, sf=48, zc=2 | 32.42 | 0.90 | 28.55 | 35.01 | 0.943 | 0.009 | 0.911 | 0.963 |
| d=4, sf=32, zc=1 | 32.85 | 0.93 | 29.11 | 35.29 | 0.950 | 0.008 | 0.921 | 0.967 |
| d=4, sf=40, zc=2 | **33.83** | 1.13 | 29.08 | 36.70 | **0.958** | 0.008 | 0.924 | 0.974 |
| d=4, sf=32, zc=2 | 33.46 | 1.12 | 28.90 | 36.58 | 0.955 | 0.009 | 0.921 | 0.972 |
| d=4, sf=32, zc=3 | 33.36 | 1.11 | 28.86 | 36.61 | 0.955 | 0.009 | 0.922 | 0.972 |
| d=5, sf=32, zc=1 | 33.11 | 1.02 | 29.19 | 36.54 | 0.954 | 0.008 | 0.925 | 0.970 |
| d=5, sf=32, zc=2 | **33.93** | 0.99 | 29.31 | 36.57 | **0.959** | 0.007 | 0.924 | 0.973 |
| d=6, sf=32, zc=3 | 32.55 | 0.96 | 28.56 | 35.62 | 0.947 | 0.009 | 0.917 | 0.967 |

Table S8: **Effect of p range on 10-shot inference.** The results demonstrate that multi-shot inference improves reconstruction quality. Within a limited number of trials, the p range within [0.3, 0.7] produced the best results.

| p range | Inference type | PSNR | std | min | max | SSIM | std | min | max |
|---|---|---|---|---|---|---|---|---|---|
| | 0.5 x 10 | 34.00 | 1.61 | 28.39 | 37.61 | 0.959 | 0.011 | 0.904 | 0.974 |
| [0.5, 0.5] | 0.5 x 2 (same split) | 34.00 | 1.62 | 28.35 | 37.52 | 0.959 | 0.011 | 0.903 | 0.974 |
| | 0.5 x 1 | 33.49 | 1.57 | 28.31 | 37.28 | 0.954 | 0.011 | 0.900 | 0.970 |
| | 0.7 x 10 | 34.35 | 1.72 | 29.49 | 38.19 | 0.961 | 0.010 | 0.919 | 0.975 |
| [0.3, 0.7] | 0.7 x 10 (median) | 34.33 | 1.72 | 29.48 | 38.16 | 0.961 | 0.010 | 0.919 | 0.974 |
| | 0.7 + 0.3 | 34.24 | 1.72 | 29.49 | 38.16 | 0.961 | 0.010 | 0.919 | 0.975 |
| | 0.7 x 1 | 34.08 | 1.68 | 29.22 | 37.83 | 0.958 | 0.010 | 0.914 | 0.973 |
| [0.25, 0.75] | 0.75x 10 | 34.34 | 1.20 | 29.33 | 36.96 | 0.960 | 0.007 | 0.927 | 0.975 |
| [0.2, 0.8] | 0.8 x 10 | 34.09 | 1.18 | 29.08 | 36.71 | 0.959 | 0.008 | 0.921 | 0.974 |

# C   Appendix / Supplementary figures

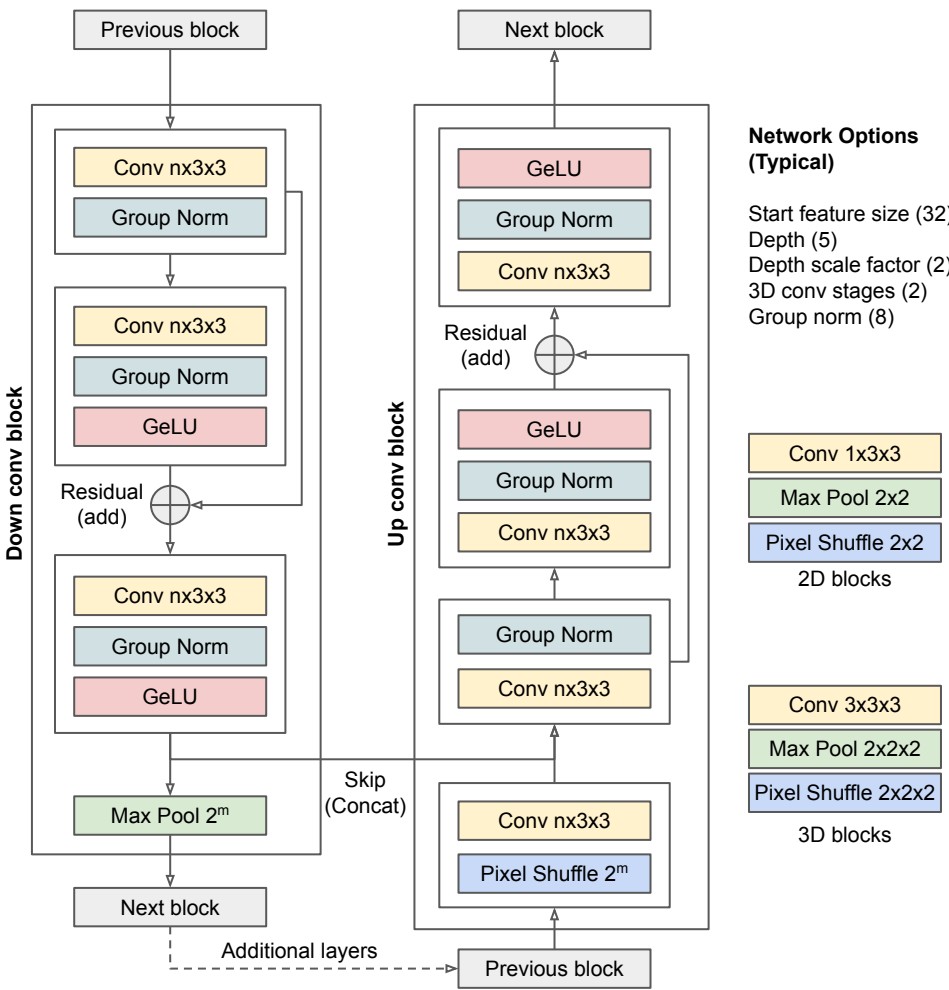

Figure S1: **The network diagram showing the primary ResUNet components used in this work.** The down convolution block has 3 convolutions, with a residual connection between the first two layers. The up-convolution block has a similar structure with the up-sampling performed using a pixel shuffle layer followed by a convolutional layer.

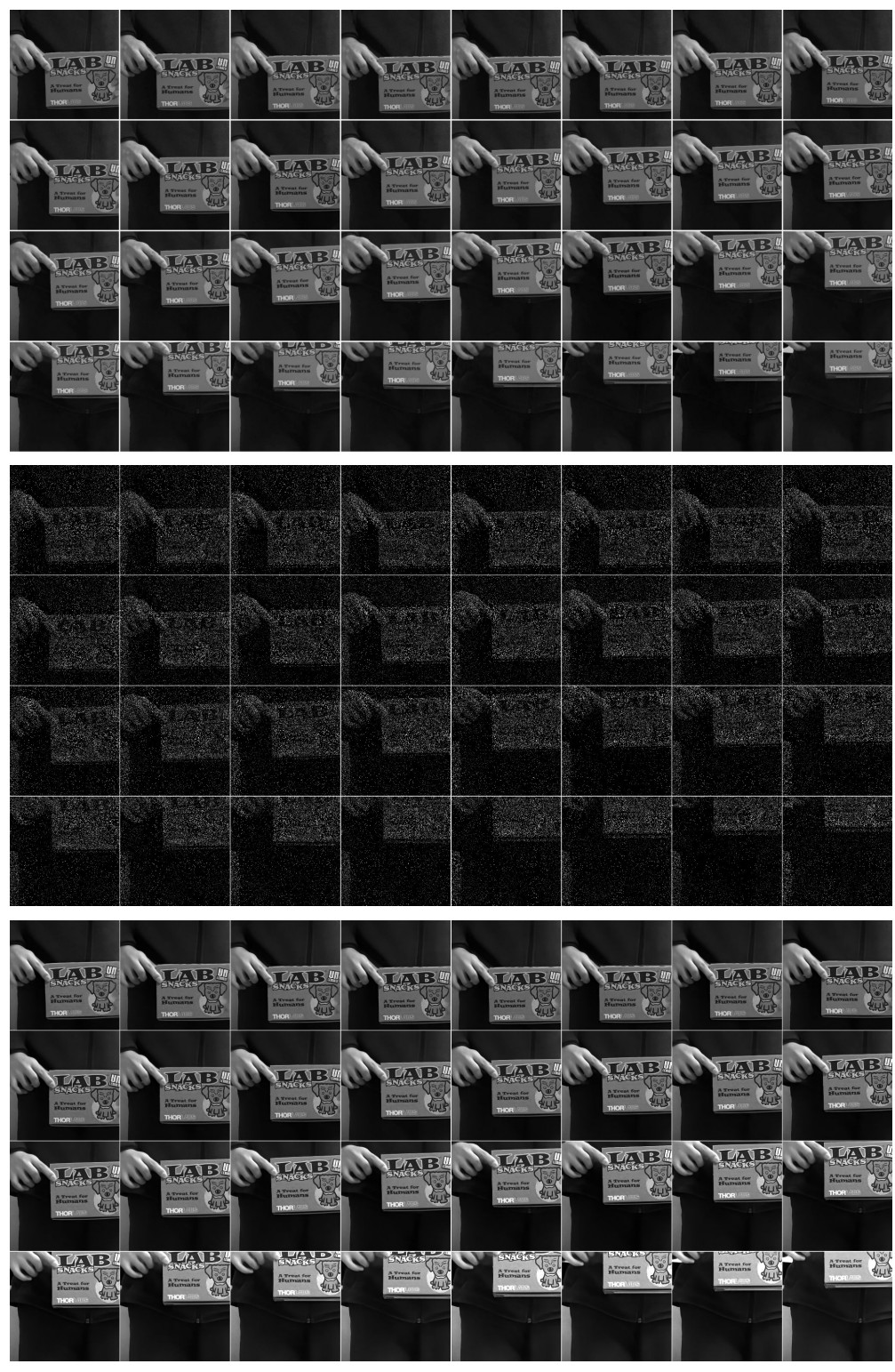

Figure S2: **Example key frames.** Top: The ground truth. Middle: The simulated 1-bit quanta data. Bottom: our reconstruction.

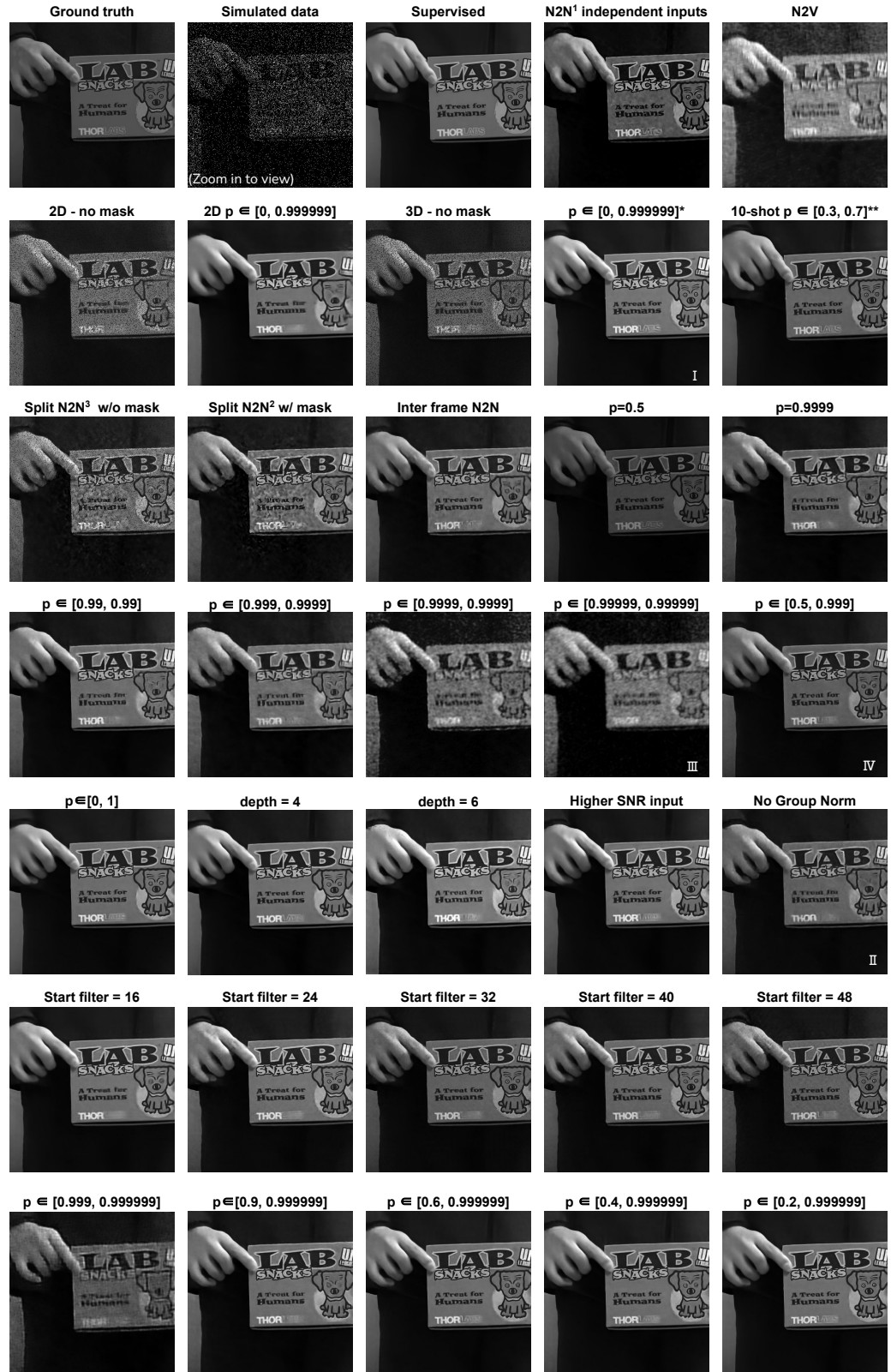

Figure S3: **Visualization of selected training experiments under conditions indicated by the title.** The Rome numbers correspond to experiments in Fig. 5. The hyperparameters and the PSNR/SSIM of most of these images and additional experiments are provided in the HP.CSV in the supplemental materials for reference. (Zoom in for detail)

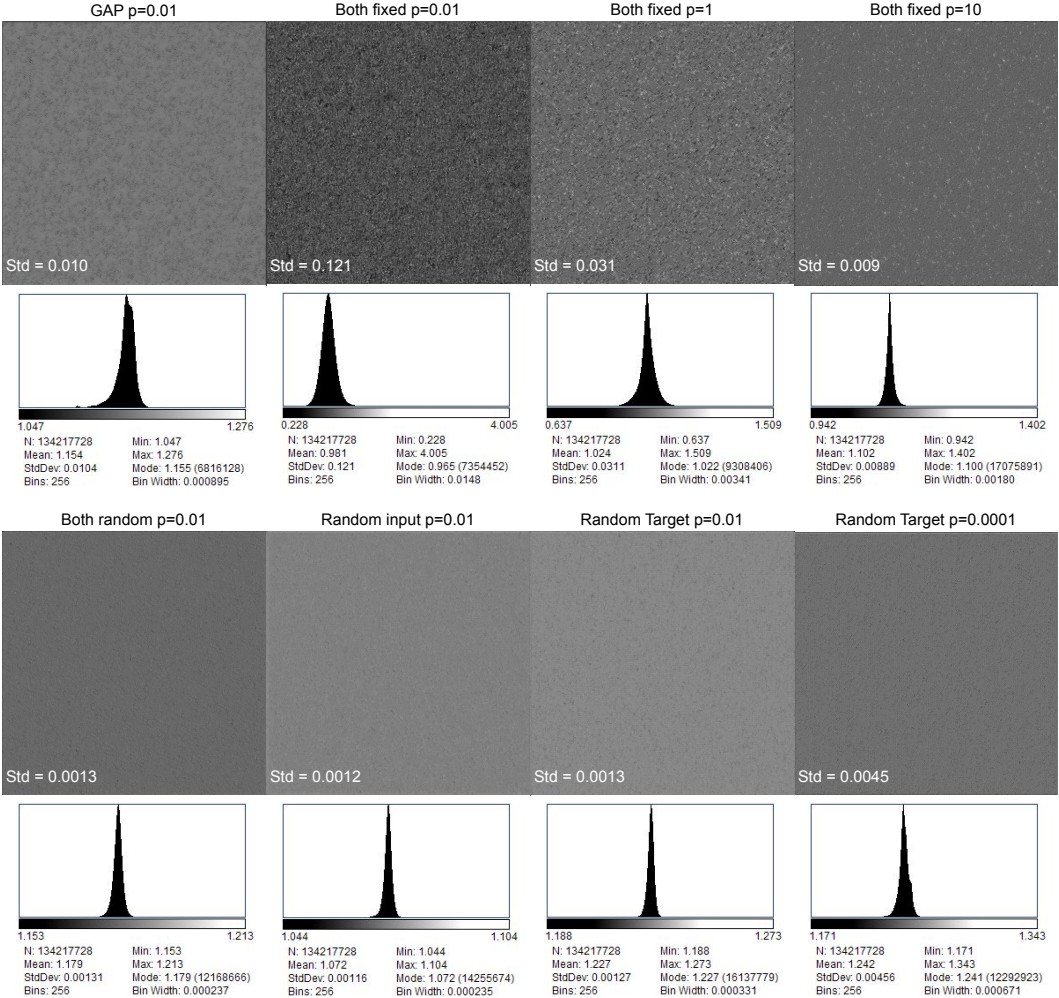

Figure S4: **Results from fixed and randomized training pairs of Poisson random noise.** An ideal training method and model should produce a pure blank image with a low standard deviation. Top Row: The input is randomly generated noise sampled from a white image using a constant Poisson rate $p$. In GAP, a large fixed thinning parameter (0.99) is used to split only a few photons to the target, resulting in mostly blank images with a few random photons. In all cases on the top row, the granular pattern is obvious in the image. Bottom Row: The problem is less prominent when either or both input and target are random. Reducing the number of photons in the target by 100 times increases the standard deviation but is not as significant as seen in the cases above.

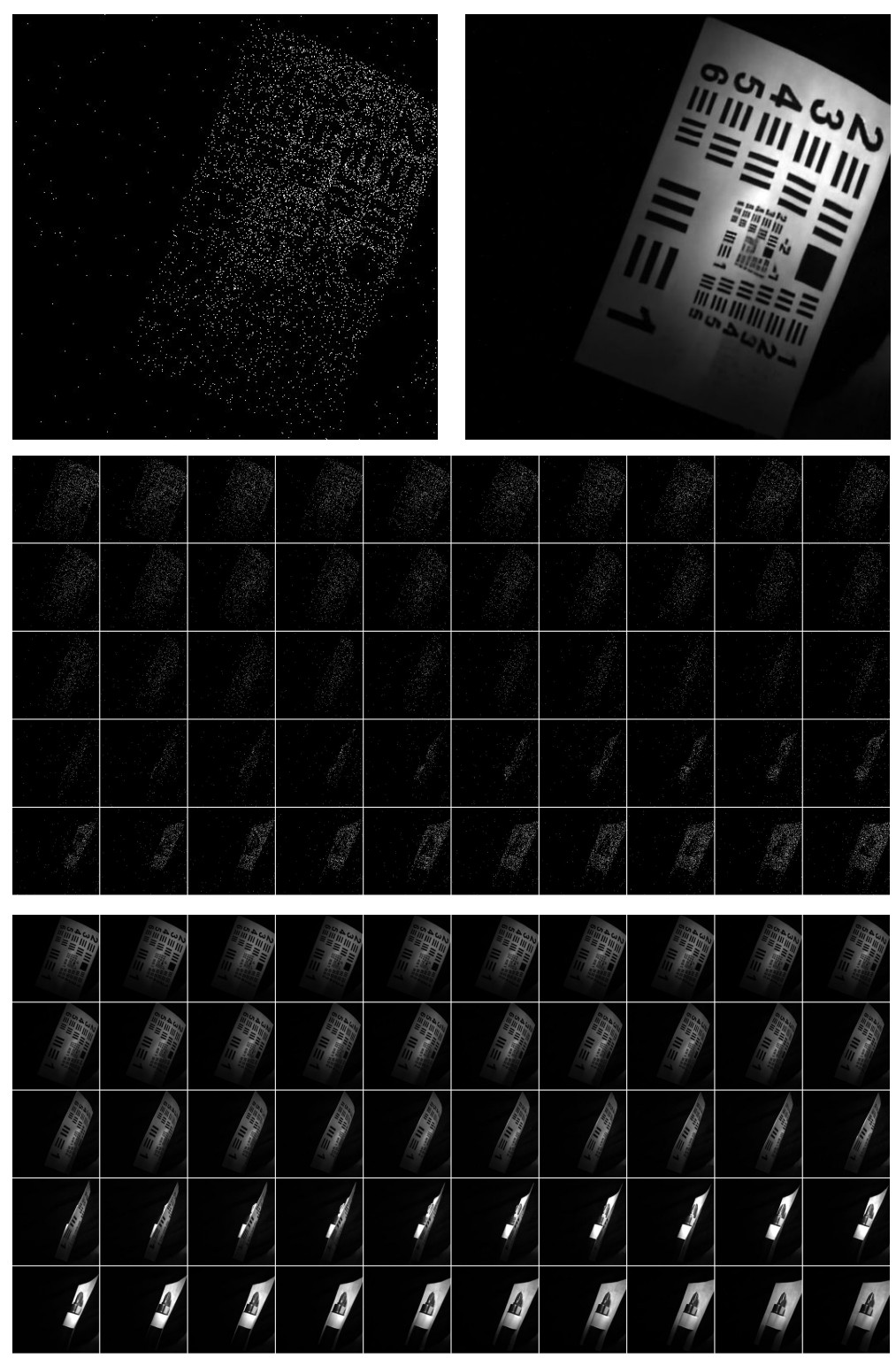

Figure S5: **Example real SPAD data and reconstruction: Resolution target rotating on a drill bit.** This scene shows a USAF 1951 resolution target spinning in a dark room. The object is rotating. Top right: the raw data. Middle: 50 frames of raw data, skipping 50 frames. Bottom: the corresponding reconstruction.

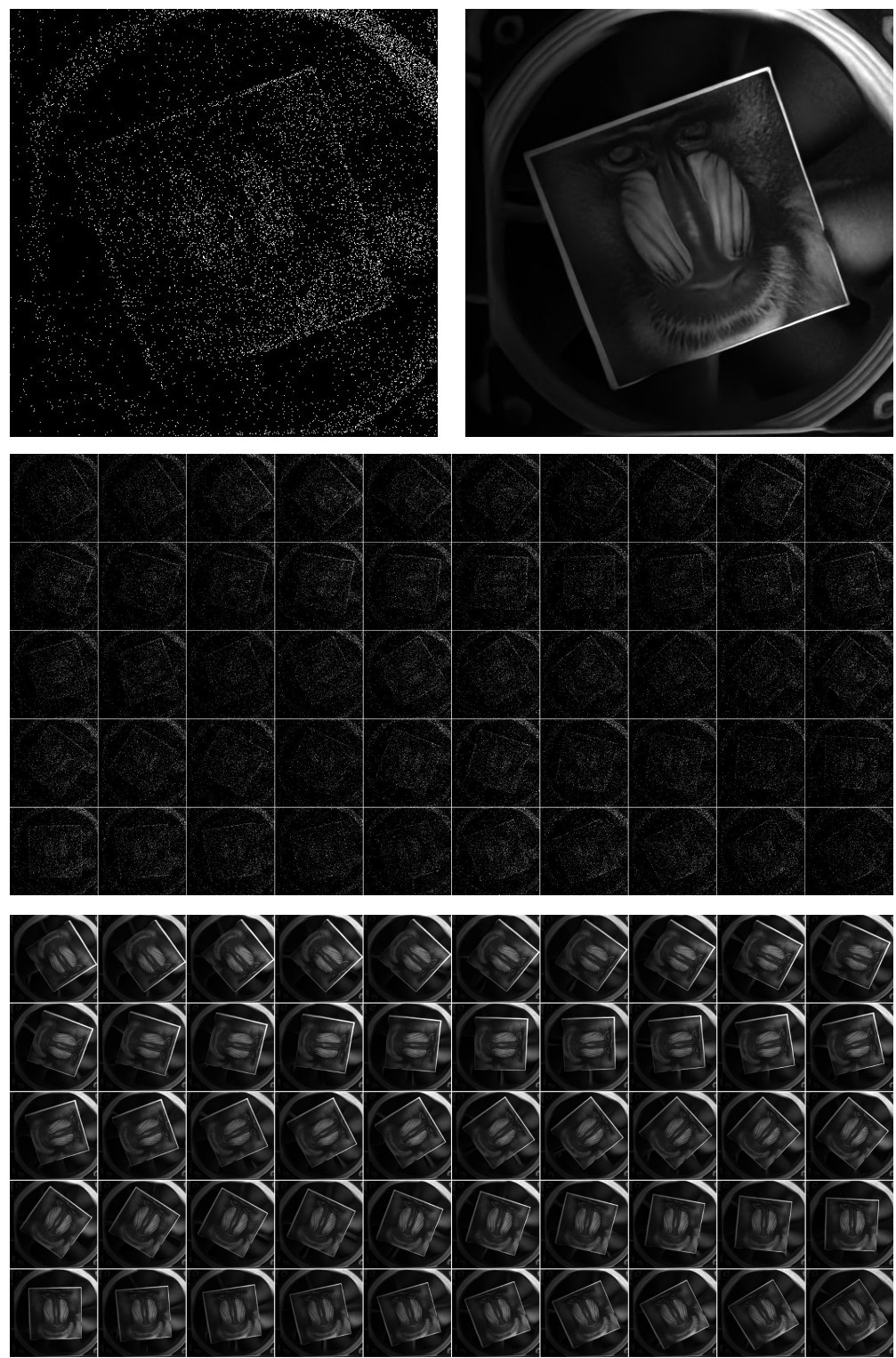

Figure S6: **Example real SPAD data and reconstruction: Image rotating on the CPU fan.** This scene is shown in Fig 1, showing a sticker of a mandrill image rotating on a CPU fan acquired in a dark room. The scene is in low light. The camera is static. Part of the scene is rotating. The speed of the CPU fan is 1500 RPM. The light intensity of the scene is in the order of 1-10 lux. This is a scene in a dark room with the room light turned off, with the only light source the computer monitor pointing towards the wall and some small LEDs on the motherboard. It is worth noting that even at this low light condition, the imaging is not photon-limited because the hardware is highly sensitive. We had to reduce the aperture of the camera lens to ensure the sensor was not over-saturated. Top left: The raw data. Top right: the reconstruction. Middle: 50 frames of raw data, skipping 50 frames. Bottom: the corresponding reconstruction.

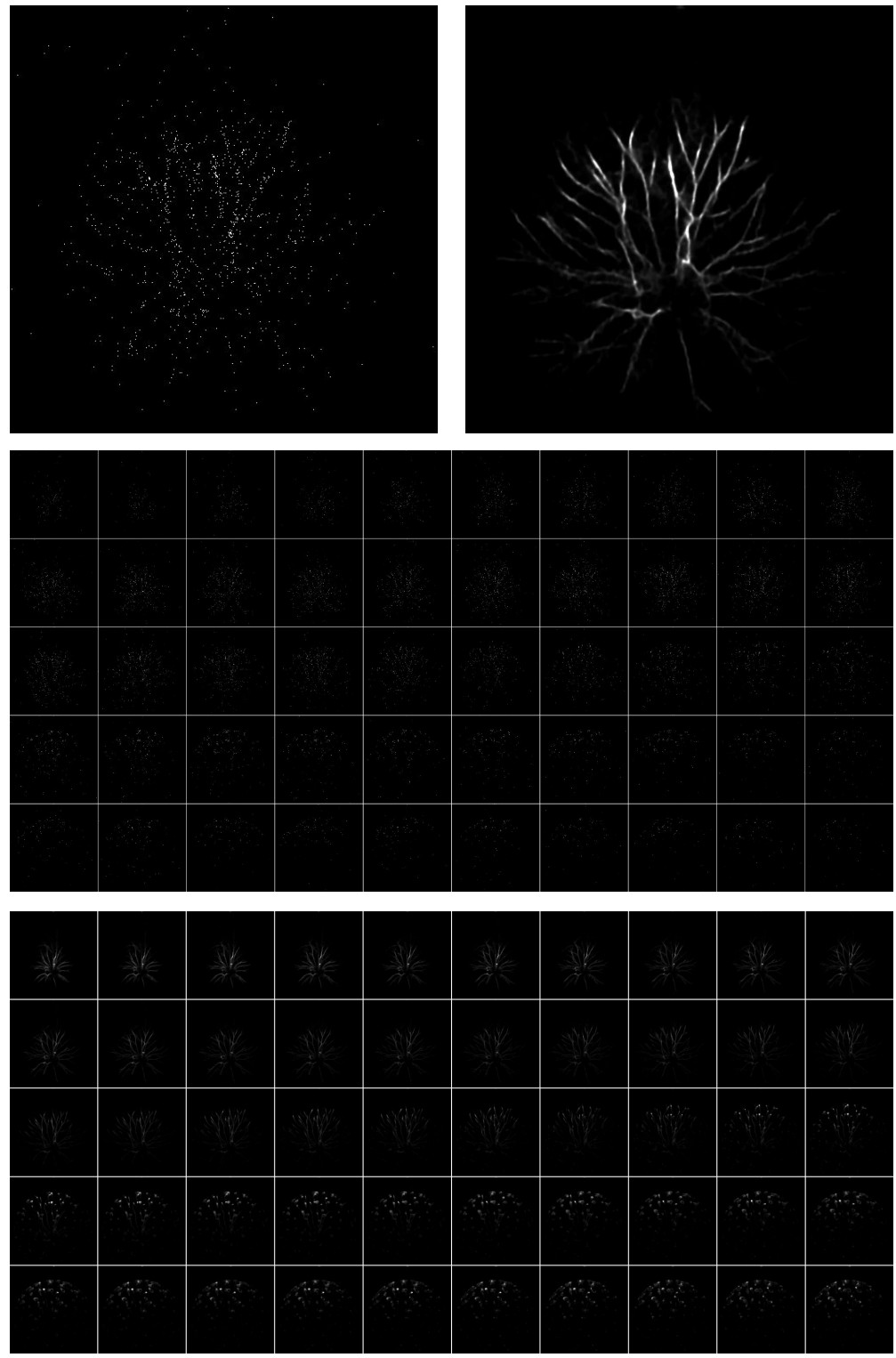

Figure S7: **Example real SPAD data and reconstruction: Plasma ball.** This scene shows a plasma ball, which produces plasma in a vacuum sphere. Plasma is triggered by high voltage generated through a buck converter circuit, with a measured frequency of 28 kHz. The plasma is released in a pulsed fashion at this frequency, following similar paths between adjacent events. The camera is triggered at a similar frequency to capture each image, representing a 6 ns snapshot of the event. Adjacent frames indicate the flow of the plasma, capturing this photo-sparse scene with extremely high-speed events. Top left: The raw data. Top right: the reconstruction. Middle: 50 frames of raw data, skipping 500 frames. Bottom: the corresponding reconstruction.

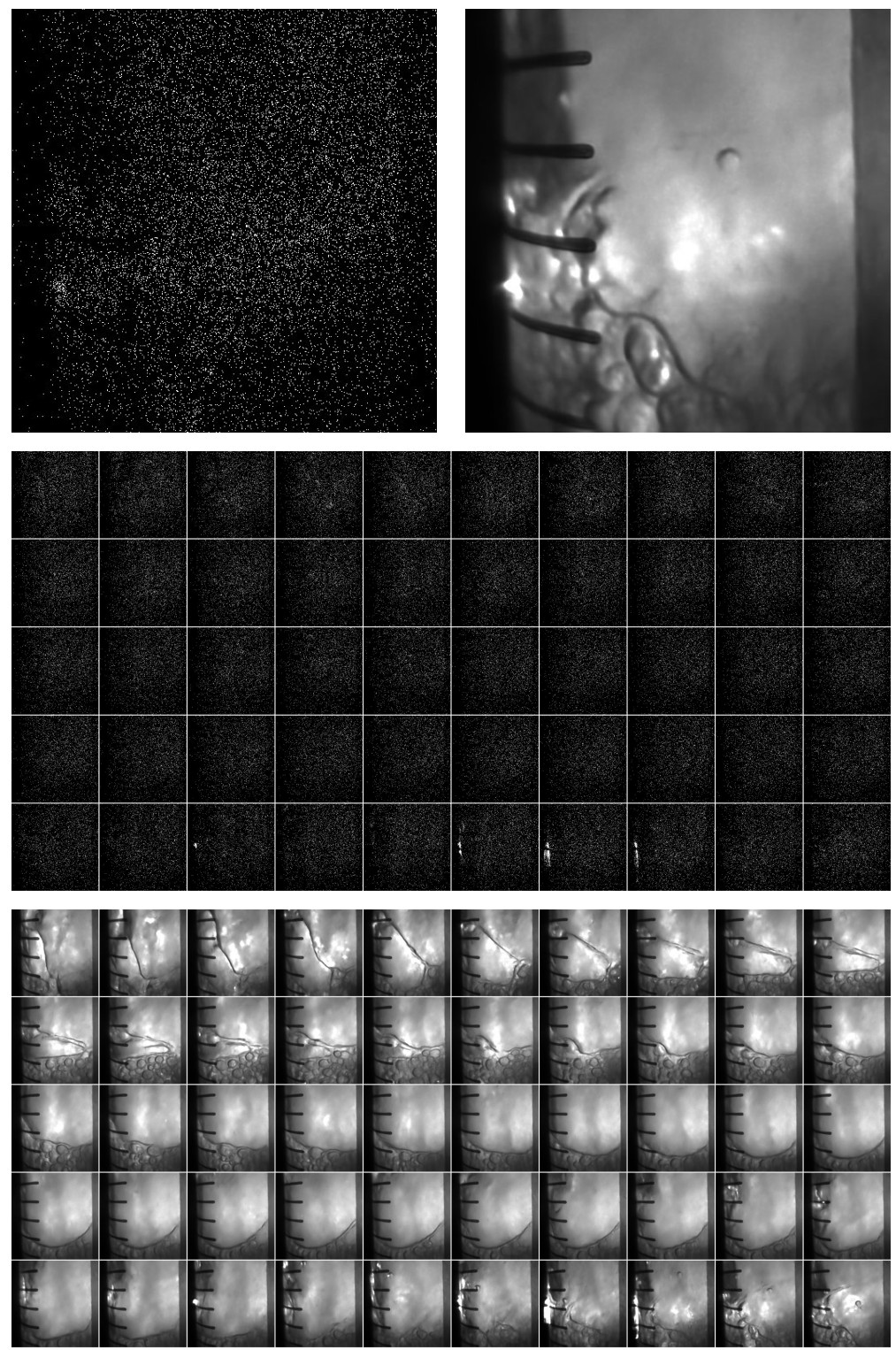

Figure S8: **Example real SPAD data and reconstruction: Sonicator and bubbles.** This scene shows a piezo transducer sending a 3 MHz sound wave through a detergent liquid, creating bubbles, mist, and water droplets. It is a highly dynamic, complex, and chaotic scene with random high-speed movement. The high background signals from the mist pose a challenge. Despite this, our method demonstrated reasonable performance. Top left: The raw data. Top right: the reconstruction. Middle: 50 frames of raw data, skipping 500 frames. Bottom: the corresponding reconstruction.

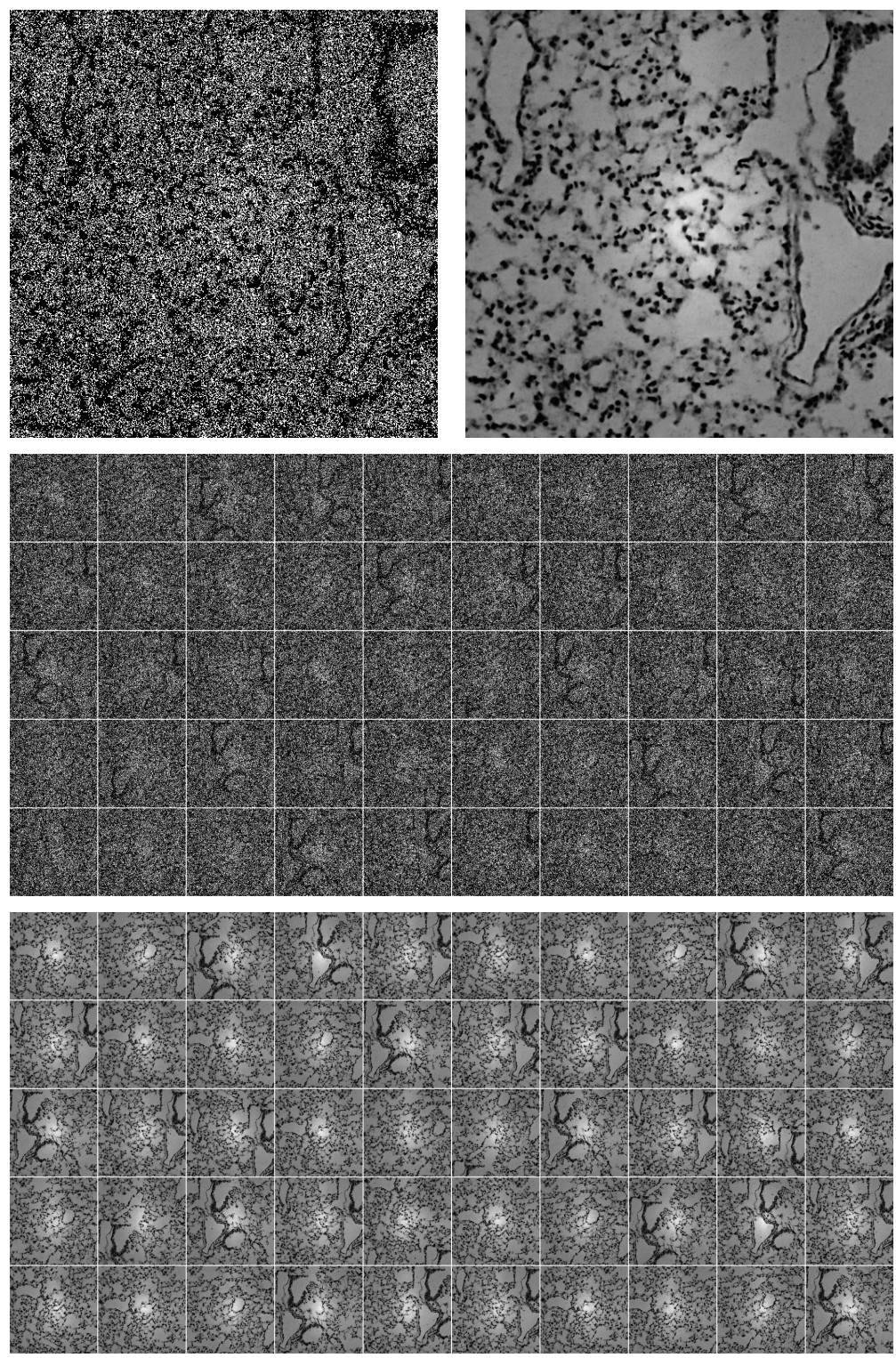

Figure S9: **Example real SPAD data and reconstruction: Moving H&E microscope slide.** This scene was taken using a bright field microscope under 20x. The sample is a H&E staining histology slide of mouse lung tissue. The sample is dynamically moved in a square pattern on a motorized stage. Most pixels are saturated in this case. Top left: The raw data. Top right: the reconstruction. Middle: 50 frames of raw data, skipping 500 frames. Bottom: the corresponding reconstruction.

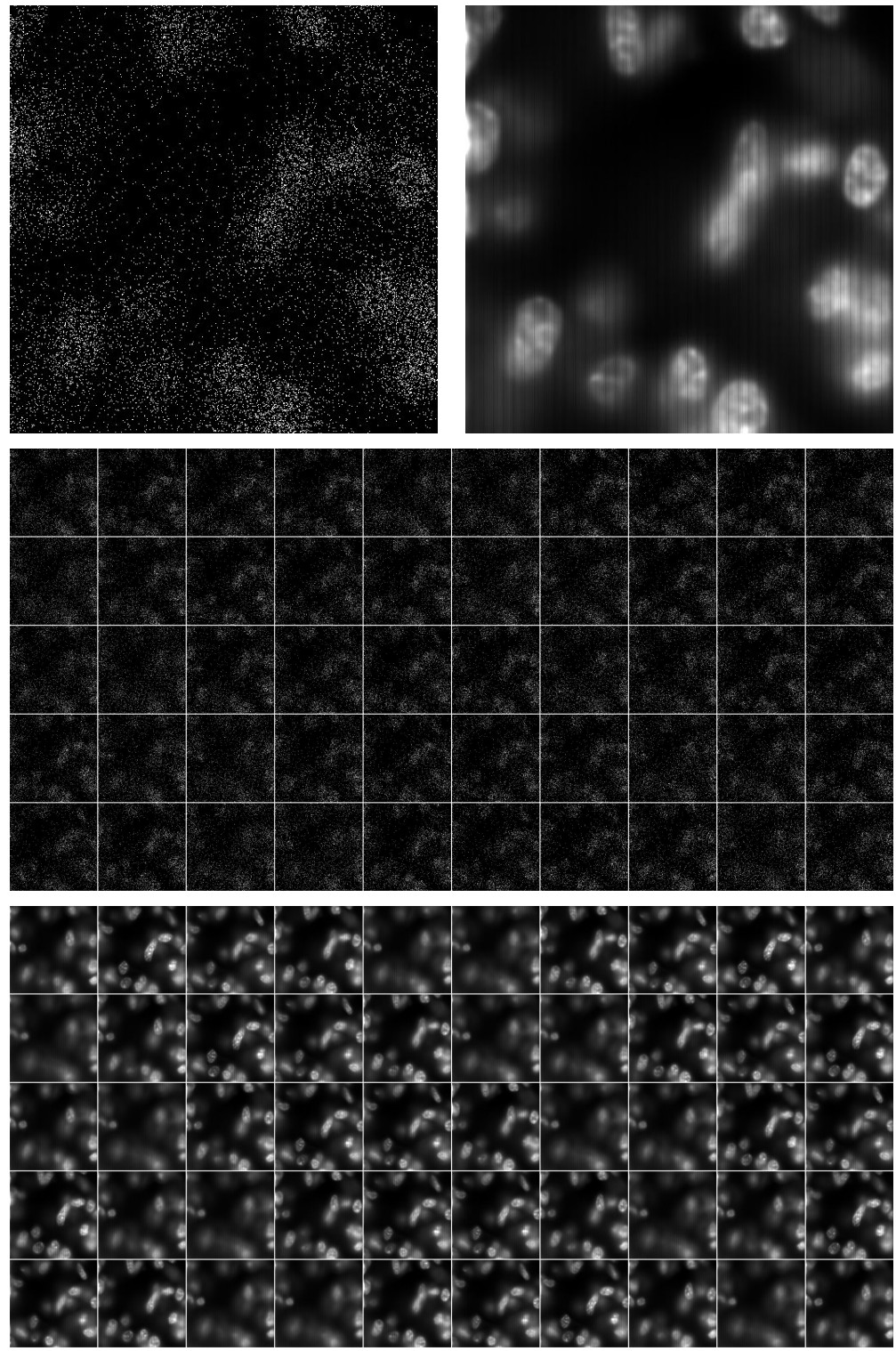

Figure S10: **Example real SPAD data and reconstruction: dynamic focusing on cells.** This scene was captured using a wide-field fluorescence microscope at 100x magnification, showing the nuclei of cultured cells. The focus of the microscope was dynamically shifted with an electronically tunable lens. A fixed pattern noise is present in this dataset, potentially due to hardware issues. While the pattern is not obvious in a single binary frame, it becomes clearly visible after reconstruction at a single-pixel width. This is a relatively low-light condition with dynamic motion. Top left: The raw data. Top right: the reconstruction. Middle: 50 frames of raw data, skipping 500 frames. Bottom: the corresponding reconstruction.

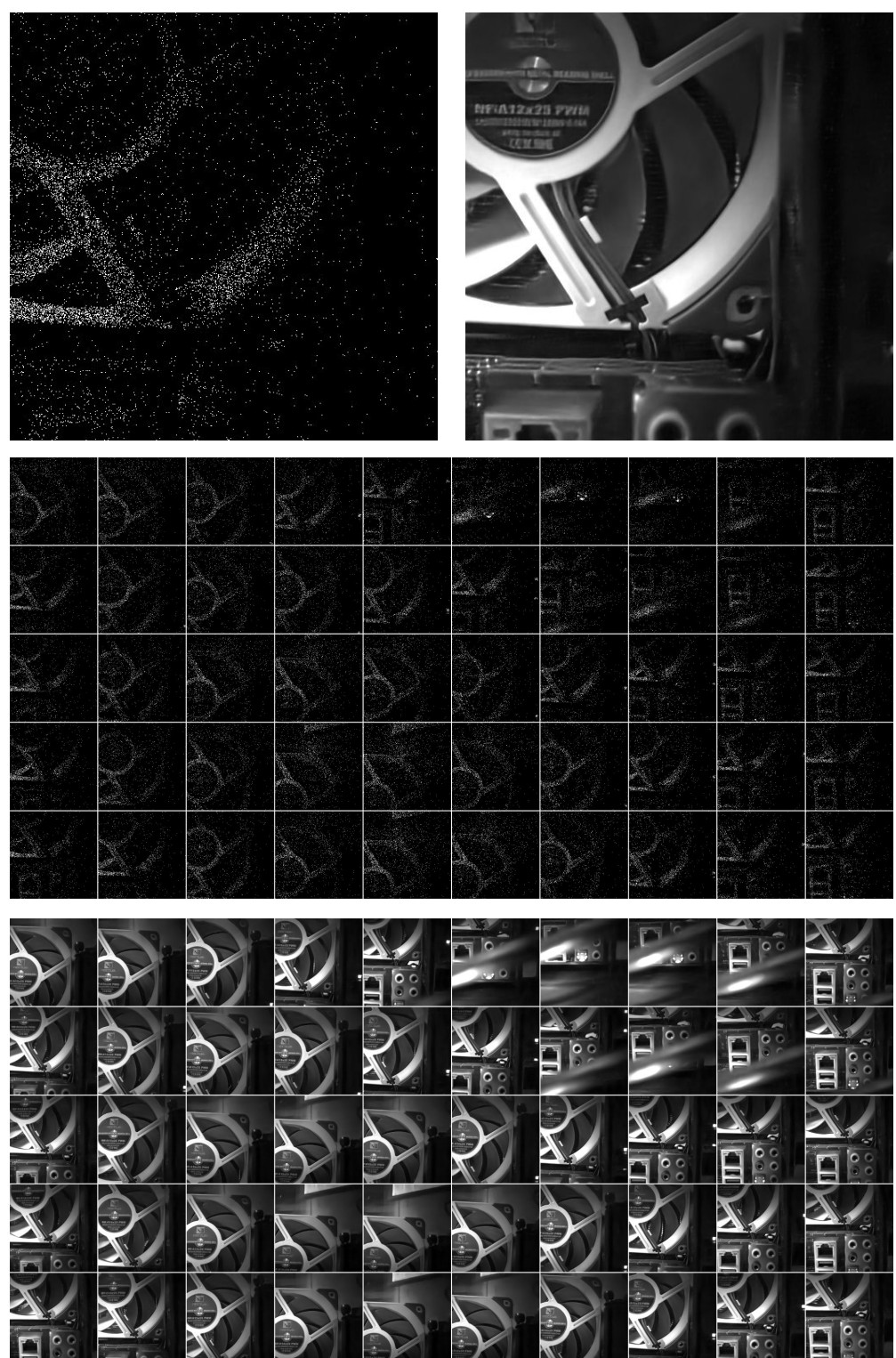

Figure S11: **Example real SPAD data and reconstruction: CPU fan with camera motion.** This scene was taken in a dark room, showing a running CPU fan. The exposure was 6 ns for each image. The camera is moving up and down. Individual fan blades and the characters on the fan were both resolved after reconstruction. This is a very low-light scene with dynamic motion on both the object and the camera. Top left: The raw data. Top right: the reconstruction. Middle: 50 frames of raw data, skipping 500 frames. Bottom: the corresponding reconstruction.

Model applied
to the data
used in
training

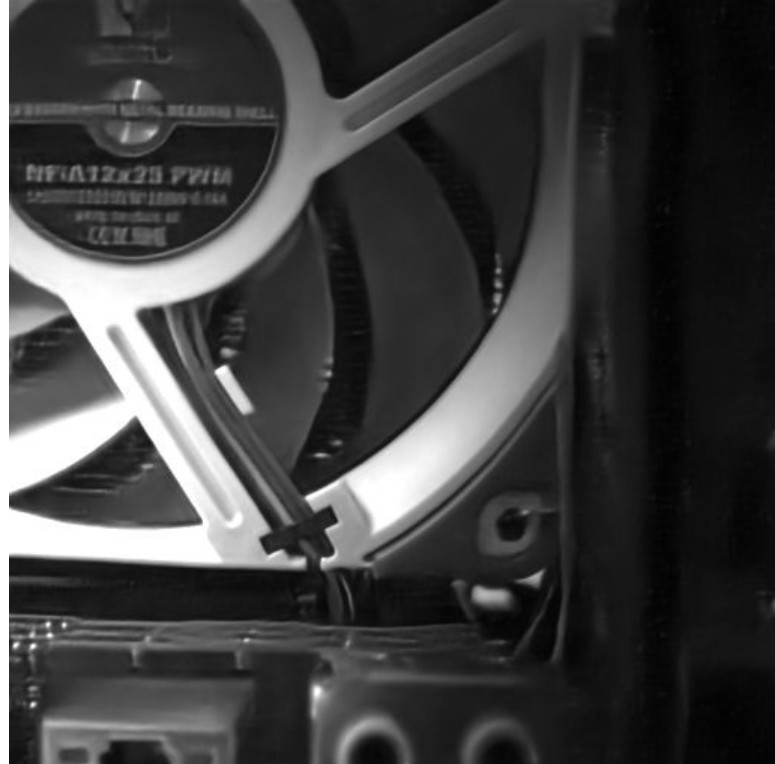

Model applied
to data from a
similar scene
but **NOT** used
in training

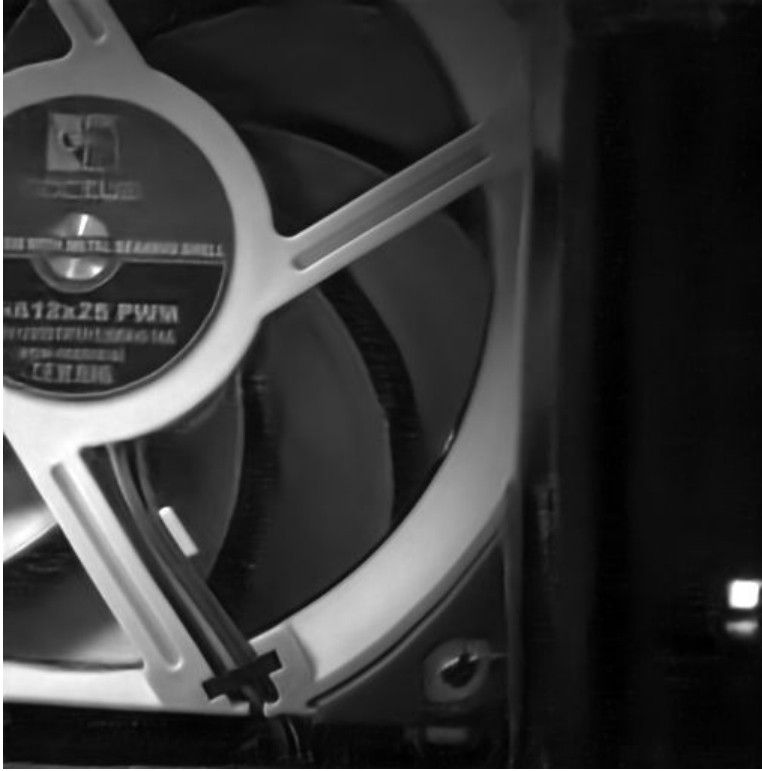

Figure S12: **The result from a model applied to data not used in the training.** The model used in SF11 was applied to data taken in the same scene. The data used in inference was not used in the training. No qualitative difference was noticed.

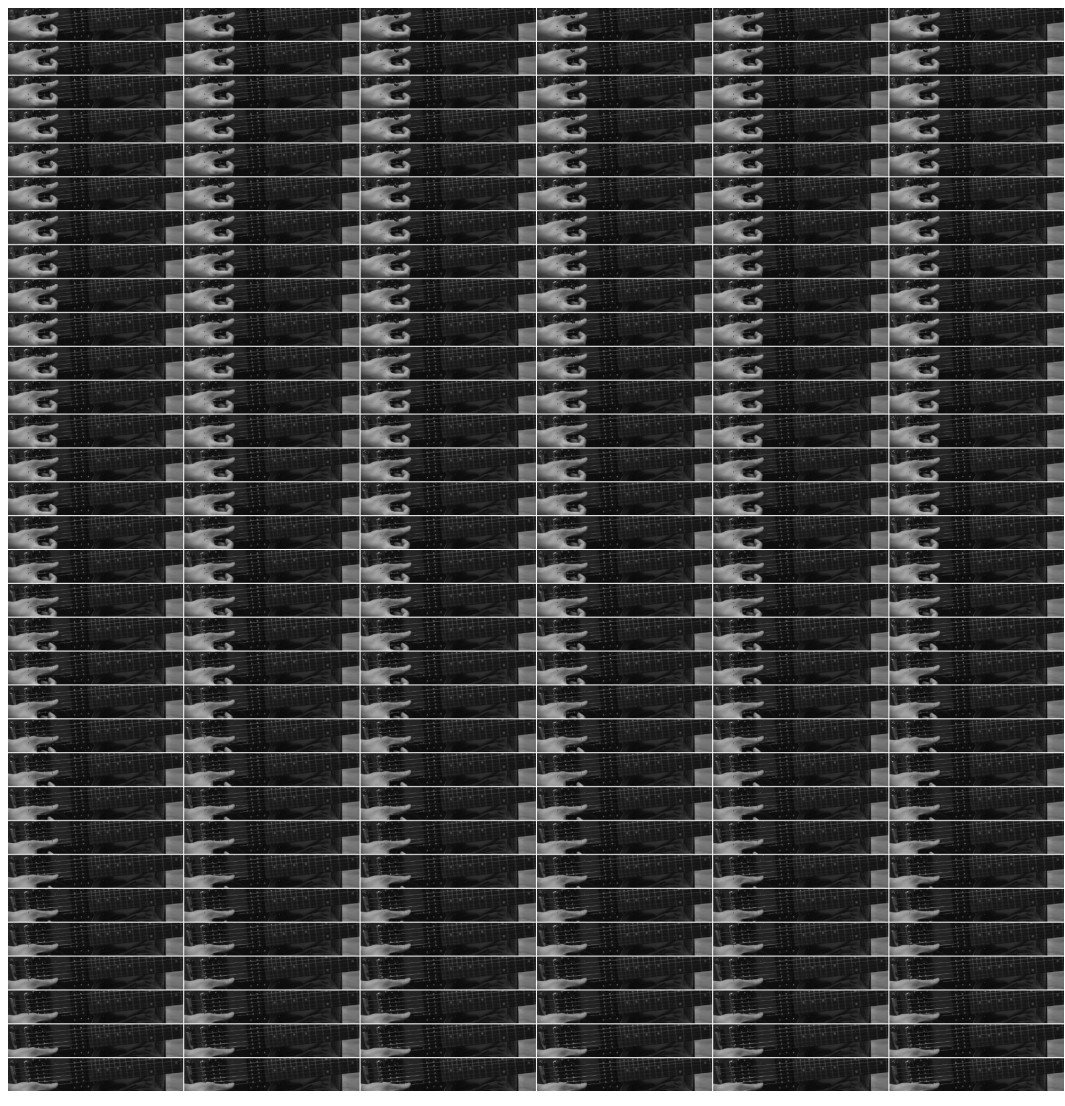

Figure S13: **Selected crops of selected keyframes from our reconstructions in Fig. 6.** Showing the consistent quality and the movement of the hand.

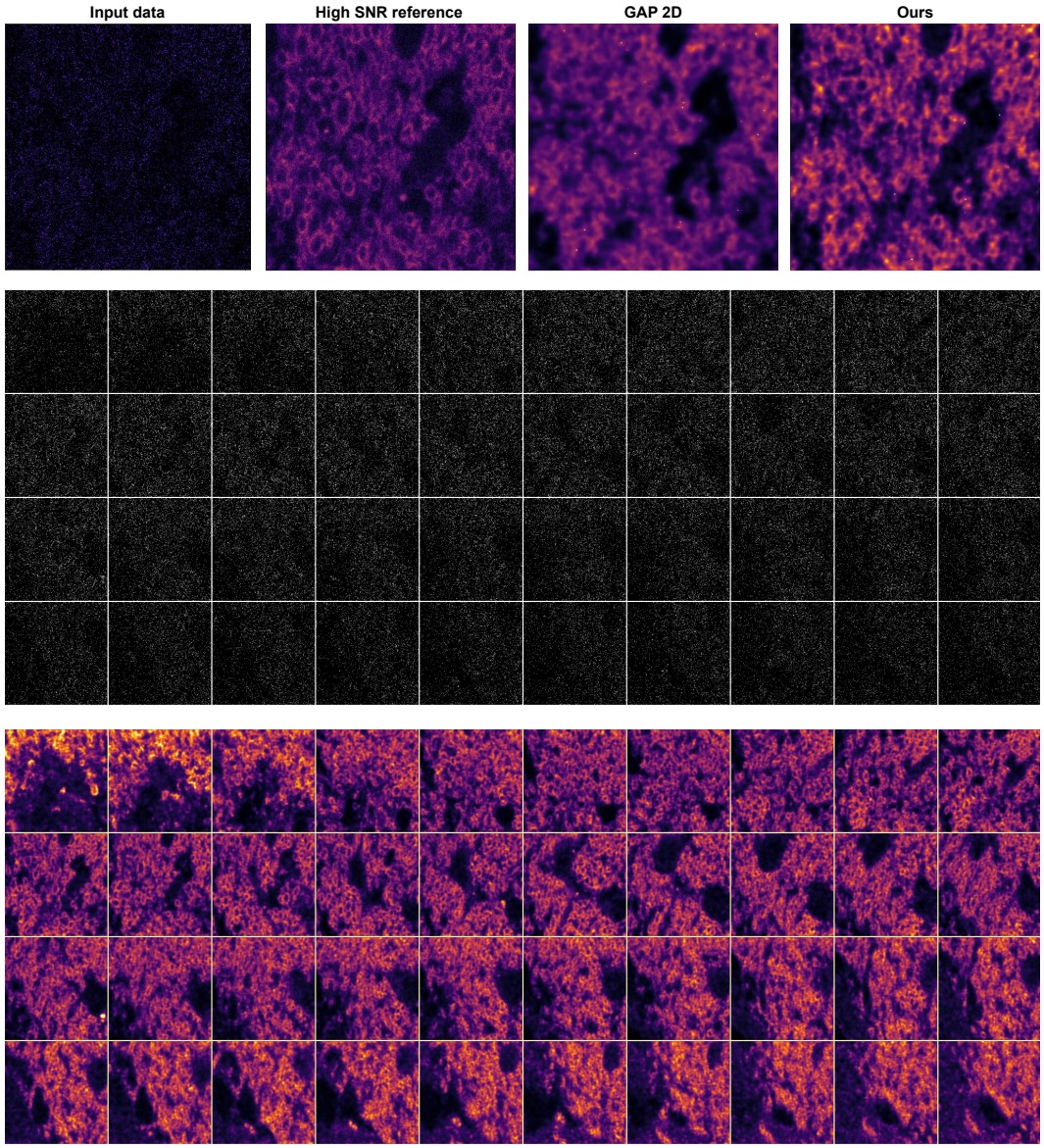

Figure S14: **Example results from photon-sparse confocal 3D volume.** The data was acquired using a Leica SP8 confocal microscope in photon counting mode, showing the auto-fluorescence of a mouse brain tissue. Input data, high SNR reference, the result from the original GAP open source code, and our results are shown on the top row for qualitative comparison. The high SNR reference is scanned separately in a different imaging session. There are small differences between the images due to repositioning. Thus, no numerical comparison is available. Raw data and our reconstruction of key z-slices through the volume are shown below.

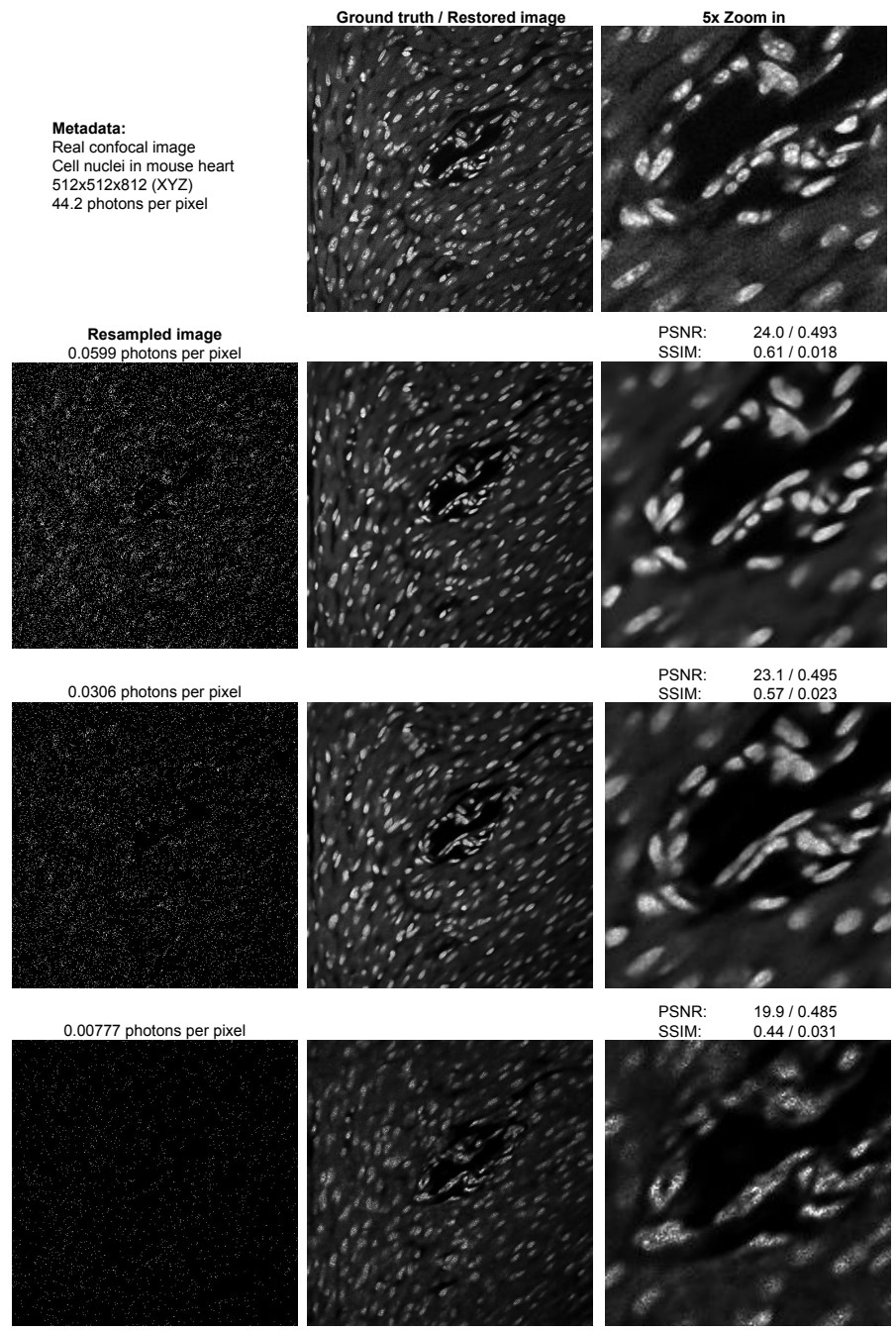

Figure S15: **Example results from simulated photon-sparse confocal 3D volume.** The data was acquired using a Leica SP8 confocal microscope in normal mode, showing the DAPI-stained cell nuclei in mouse heart tissue. The data was then sampled into different levels of photon counts and truncated at 1 to simulate the SPAD data. PSNR and SSIM are computed for different input photon levels.

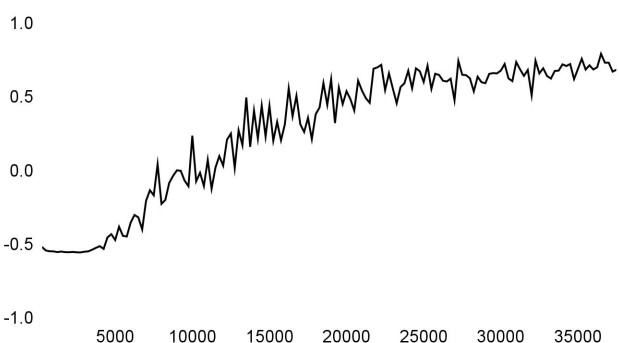

Figure S16: **Example learning curve from N2N training.** The validation loss increases since a very early stage of the training. The X-axis indicates trainning steps.

# D   Appendix / acronyms

**SPAD**  single-photon avalanche diode

**QIS**  quanta image sensor

**CV**  computer vision

**CNN**  convolution neural networks

**MSE**  mean squared error

**MMSE**  minimal mean squared error

**GAP**  Generative Accumulation of Photons

**N2N**  Noise2Noise

**N2V**  Noise2Void

**N2S**  Noise2Self

