# OpenReview forum: "bit2bit: 1-bit quanta video reconstruction via self-supervised photon prediction"
_NeurIPS.cc/2024/Conference — NeurIPS 2024 poster_

### Official Review · Reviewer_VmGM · 2024-07-13

**Soundness:** 3
**Presentation:** 3
**Contribution:** 4
**Rating:** 6
**Confidence:** 5

**Summary:**

1. The paper introduces a novel self-supervised method for reconstructing high-quality image sequences from sparse binary quanta image data.

2. The paper mainly adapt a self supervised denoising algorithm called GAP. Instead of directly adopting the GAP method, the authors extended it to spatiotemporal structures and proposed a novel masked loss to address the correlation issue between input and target images.

3. Experimental results demonstrate that the proposed method substantially outperforms state-of-the-art techniques such as Quanta Burst Photography (QBP) in both reconstruction quality and throughput.

4. Additionally, the paper presents a new dataset and discusses the potential of the method for generalizing to other spatial event point processes beyond the specific application of quanta image sensors.

**Strengths:**

1. The strengths of this approach mainly lie in its ability to effectively utilize spatiotemporal information, its novel masked loss function that addresses the correlation between input and target images, and its demonstrated superiority over existing methods like Quanta Burst Photography in terms of reconstruction quality and throughput efficiency.

2. In addition, as a self supervised algorithm, the training of this scheme does not require the construction of large-scale synthetic datasets, making it more flexible and convenient.

**Weaknesses:**

1. In Equation 1, there is a confusion between x_{in} and x_{inp}. Moreover, the right side of Equation 1 seems to have overlooked x_{tar}.

2. To illustrate the application of QIS in high-speed and low-light scenarios, it would be beneficial for the authors to provide specific data on the rotation speed of the fan  and the light intensity of the scene.

3. The authors used a 10-shot approach to calculate the average results. However, methods similar to GAP require many iterations during inference, and the combination of these two factors could lead to significant computational time. The authors should provide more detailed explanations, such as the number of iterations and the computational workload involved.

**Questions:**

1. I still have doubts about the role of the masked loss. The authors state that this loss is intended to address the correlation issue between the input and target images. However, after convolution, the information from locations that were originally 1 in the input image has already spread to other pixel locations. Simply masking the original 1 positions may not fully conceal the information from those positions.

2. I am not quite clear about what the network's output is, given that the input is a t*h*w matrix. Is the output also t*h*w? If so, in the supervised loss comparison experiments, are there 32 grayscale images gt? For the N2N comparison experiment, is the supervision using QIS data with a shape of t*h*w? How was the GAP-2D experiment designed—does the network input consist of the first frame of QIS data from t*h*w?

3. Why does N2N lead to a granular pattern? Why can GAP-like method resolve the granular pattern caused by the N2N method?

A good response to the Weaknesses and Questions will improve my initial rating.

**Limitations:**

There is no discussion of potential negative societal impacts.

---

> ### Author Rebuttal · Authors · 2024-08-05
>
> We appreciate the reviewer’s insightful comments. Here, we answer the reviewer’s questions one by one:
>
> 1. **In Equation 1, there is a confusion between x_{in} and x_{inp}. Moreover, the right side of Equation 1 seems to have overlooked x_{tar}.**\
> Thank you for pointing out the mistake. We will correct the typo in our revision. $x_{in}$ should be $x_{inp}$. There should be a $x_{tar}$ before the $\ln$ in Equation 1.
>
> 2. **To illustrate the application of QIS in high-speed and low-light scenarios, it would be beneficial for the authors to provide specific data on the rotation speed of the fan and the light intensity of the scene.**\
> Agree and we will include the information in the revision. The rotation speed of the fan is 1500 rpm. The light intensity of the scene is in the order of 1-10 lux. This is a scene in a dark room with the room light turned off, with the only light source the computer monitor pointing towards the wall and some small LEDs on the motherboard. It is worth noting that even at this low light condition, the imaging is not photon-limited because the hardware is highly sensitive. We had to reduce the aperture of the camera lens to ensure the sensor was not over-saturated.
>
> 3. **The authors used a 10-shot approach to calculate the average results. However, methods similar to GAP require many iterations during inference, and the combination of these two factors could lead to significant computational time. The authors should provide more detailed explanations, such as the number of iterations and the computational workload involved.**\
> Thank you for this comment. We notice the 10-shot approach is not clearly defined in the original manuscript. We will clarify this in L197 when the term is first introduced in the revision.
> To clarify, the 10-shot approach is unrelated to the iterative sampling in GAP. GAP uses multiple iterations to achieve posterior sampling and function as a generative model. In this work, we only use the network for MMSE denoising, which requires only a single iteration. The 10-shot refers to running the inference 10 times using data Bernoulli-resampled from the raw data (L246). This takes 10 times the inference time but is not a major time-limiting factor. If the data throughput is a concern, we also provide a practical one-shot inference solution (L166, L195, L243).
>
> 4. **I still have doubts about the role of the masked loss. The authors state that this loss is intended to address the correlation issue between the input and target images. However, after convolution, the information from locations that were originally 1 in the input image has already spread to other pixel locations. Simply masking the original 1 positions may not fully conceal the information from those positions.**\
> The purpose of the masking is not to fully conceal the information from these positions. The only input information to the network is photon positions, so it is necessary for the network to spread this information to other pixels to make predictions about the clean signal. Due to the binary nature of the sensor and the splitting of the truncated Poisson (Bernoulli) distributed data, pixels that have the value 1 in the input will always have the value 0 in the target. Training a network without masking will lead to the network predicting dark pixels at the locations where there is a photon in the input (see figure in the rebuttal PDF). The mask prevents the network from learning this deterministic relationship by zeroing the loss computation for pixel locations that have value 1 in the input. Other (neighboring) locations are not affected by this issue and do not need to be masked. The mask prevents the network from learning from the target pixels that are determined to be 0 because the input is 1.
>
> 5. **I am not quite clear about what the network's output is, given that the input is a thw matrix. Is the output also *thw*? If so, in the supervised loss comparison experiments, are there 32 grayscale images gt? For the N2N comparison experiment, is the supervision using QIS data with a shape of *thw*? How was the GAP-2D experiment designed—does the network input consist of the first frame of QIS data from *thw*?**\
> Correct. The network's input and output have the same *thw* dimension (3D to 3D). There are 32 grayscale ground truth frames in the supervised loss comparison experiments. The shape of the training data was always the same *thw* for all 3D network experiments, including the N2N comparison experiments. The GAP-2D experiment is designed so that individual 2D frames are used in training and inference. The network input consists of single frames of the QIS data, randomly selected during training (random *t* from *thw*).
>
> 6. **Why does N2N lead to a granular pattern? Why can GAP-like method resolve the granular pattern caused by the N2N method?**\
> We believe the granular pattern is caused by overfitting due to limited training data. In N2N, the same binary data pairs are repeatedly used for training. In our method, random splitting alleviates this problem by creating a different binary pair each time. We see this as a type of data augmentation that effectively increases the amount of available training data. L266 has some discussion regarding this. L275 and Fig. S4 demonstrate some aspects of the problem with an experiment using random Poisson noise.
>
> 7. **There is no discussion of potential negative societal impacts.**\
> We will clarify potential negative societal impacts in our revised conclusion. The method indeed can be misused in research (e.g., using inappropriate data, making wrong assumptions about the noise, etc.) and the prediction results can be misinterpreted, leading to incorrect scientific conclusions and potential negative societal impact.

---

> > ### Comment · Reviewer_VmGM · 2024-08-13
> >
> > Thank you for the kind reply, most of my concerns have been addressed. I am willing to raise the score to match the contribution of this paper.

---

### Official Review · Reviewer_iaWM · 2024-07-13

**Soundness:** 3
**Presentation:** 2
**Contribution:** 3
**Rating:** 5
**Confidence:** 4

**Summary:**

This paper presents a self-supervised method for reconstructing high-quality video from sparse binary quanta image data produced by single-photon avalanche diode (SPAD) arrays. The authors propose a novel masking strategy to handle the binary nature of the data and extend their method to 3D to leverage spatiotemporal information. They evaluate their approach on both simulated and real SPAD data, demonstrating improved reconstruction quality and throughput compared to existing methods like Quanta Burst Photography (QBP). The paper also introduces a new dataset of real SPAD high-speed videos under various challenging imaging conditions.

**Strengths:**

1.	The paper addresses an important problem in computational imaging, proposing a novel self-supervised approach for reconstructing high-quality video from sparse binary quanta data.
2.	The authors' masking strategy to handle binary data is innovative and appears effective based on the presented results.
3.	The introduction of a new real SPAD high-speed video dataset is valuable for the research community.

**Weaknesses:**

The paper fails to convincingly demonstrate the effectiveness and advantages of the proposed method over existing approaches. The theoretical foundations are not well-developed, and the experimental results are not sufficiently rigorous or comprehensive to support the claims made. The writing lacks clarity in many sections, making it difficult to fully understand the proposed method and its implications.

1.	The discussion on selecting the photon splitting variable p is confusing and lacks practical considerations. The paper doesn't adequately address how to ensure correct signal reconstruction when the signal level is unknown before capture.
2.	Many crucial implementation details are absent:
-	The composition and size of the training dataset are not specified.
-	It's unclear if the quantitative results for simulated data are computed from only one video (L204).
-	The number of videos in the real test data is not mentioned.
3.	The comparison with existing quanta video reconstruction methods is limited to only QBP (proposed in 2020). There is a lack of quantitative comparisons with other existing methods.
4.	The paper lacks a detailed analysis of the method's runtime and memory requirements compared to existing approaches.
5.	More extensive ablation studies exploring the impact of various components of the method (e.g., network architecture choices, hyperparameters) are needed to provide deeper insights into the method's performance.
6.	Several technical terms and abbreviations are not properly defined or explained:
-	L1: "SPAD" is not defined on first use.
-	L15: The unit for "0.06 photons per pixel" is not specified (e.g., per frame or per second).
-	L22: "QBP" is not defined on first use.
-	L98: The meaning of s_i and i is unclear (intensity or pixel position?).
-	L112: The probability of zero photons hitting a pixel being e^{s_i} needs a detailed explanation.
-	L197: The term "10-shot inference" is not explained.
7.	Some experimental details are missing:
-	L205: More information on the iPhone 15 slow motion mode (e.g., resolution, exact frame rate) would be helpful.

**Questions:**

1.	How do you propose to select the photon splitting variable p in practice when the signal level is unknown before capture? Can you provide guidelines for selecting an optimal value?
2.	Can you provide a more comprehensive comparison with other existing quanta video reconstruction methods beyond QBP, including quantitative results?
3.	Can you provide a detailed analysis of the computational complexity and how it scales with dataset size?
4.	Could you elaborate on the runtime and memory requirements of your method compared to existing approaches?
5.	Can you provide more extensive ablation studies on the impact of various components of your method, such as network architecture choices and hyperparameters?
6.	Can you provide more details on the composition and size of the training dataset used? How many videos were used in both simulated and real tests?
7.	How does the method perform on extremely low photon count data, and what is the lower limit of photon count where the method remains effective?
8.	How does the method perform on different types of scenes or motion patterns?

**Limitations:**

The authors have adequately addressed limitations, discussing the scope of applicability to Poisson noise and computational considerations. They have also provided a thorough discussion of assumptions and potential limitations in the conclusion.

---

> ### Author Rebuttal · Authors · 2024-08-06
>
> Thank the reviewer for the comments. We noticed several factual errors in this review:
> 1. **L1: "SPAD" is not defined on first use.**\
> SPAD is defined in L1  on first use.
> 2. **L22: "QBP" is not defined on first use.**\
> QBP is defined in L21-22 on first use.
> 3. **L205: More information on the iPhone 15 slow motion mode (e.g., resolution, exact frame rate) would be helpful.**\
> The frame rate 240 fps is reported in L205. The video is cropped/resampled to 3990x512x512.
> 4. **The composition and size of the training dataset are not specified.**\
> Data size is provided in L208. Each dataset is shown and described in Fig. S5-S12. The intended meaning of the term composition is unclear.
>
> Some questions and comments are vague and not specific. We tried our best to address them. We grouped related questions and comments:
> 1. **The paper fails to convincingly demonstrate the effectiveness and advantages of the proposed method over existing approaches. The comparison with existing quanta video reconstruction methods is limited to only QBP (proposed in 2020). There is a lack of quantitative comparisons with other existing methods. Can you provide a more comprehensive comparison with other existing quanta video reconstruction methods beyond QBP, including quantitative results?**\
> To our knowledge, our self-supervised 3D binary to 3D grayscale task is novel. We are not aware of any existing work that solves this task. We hope the reviewer can specify.
> Although similar, QBP is a substantially different task, which constructs a single 2D grayscale image from many binary frames. QBP also did not show any comparable quantitative results. We compared our method to GAP, N2N, and N2V quantitatively.
> 2. **The theoretical foundations are not well-developed, and the experimental results are not sufficiently rigorous or comprehensive to support the claims made.**\
> It is unclear which claims are not supported.
> 3. **The writing lacks clarity in many sections, making it difficult to fully understand the proposed method and its implications.**\
> It is unclear which sections and specific parts of the explanation lack clarity.
> 4. **The discussion on selecting the photon splitting variable p is confusing and lacks practical considerations. Can you provide guidelines for selecting an optimal value [for p]?**\
> We extensively discussed methods for selecting p in L157, L266 and L465, plus Fig. 3, 4, S3, S4, S6, S8, including two best practices. The simple method is using a fixed p (L160). The more robust method is selecting p randomly between 0-1 for each training pair (L166). Both methods are easy to implement in practice.
> 5. **Not adequately address how to ensure correct signal reconstruction when the signal level is unknown before capture. How to select the photon splitting variable p in practice when the signal level is unknown before capture?**\
> Our method is self-supervised. The model is trained on the very data to be denoised at the detected signal level. The selection of p is unrelated to the signal level, as it corresponds to the fraction of photons to split for already-known measurements.
> 6. **It's unclear if the quantitative results for simulated data are computed from only one video (L204).**\
> We stated the simulated data are computed from L204: a ground truth reference video.
> 7. **The number of videos in the real test data is not mentioned. Can you provide more details on the composition and size of the training dataset used? How many videos were used in both simulated and real tests?**\
> We reference all 7 real test data in L211. Each test is a video.
> 8. **The paper lacks a detailed analysis of the method's runtime and memory requirements compared to existing approaches. Can you provide an analysis of the computational complexity and how it scales with dataset size?**\
> We discussed practical computational requirements, optimization, performance, runtime, and VRAM in L185-L198. We compared our runtime to QBP (L294). The computational complexity is more relevant to the architecture rather than the data size.
> 9. **More extensive ablation studies exploring the impact of various components of the method (e.g., architecture, hyperparams) are needed to provide deeper insights into the method's performance. Can you provide more extensive ablation studies on the impact of various components of your method?**\
> It is unclear which specific ablation studies and hyperparameters are overlooked, as we provided substantial details (Table S1-8, Fig. 4, Fig. S1, 3, supp hp.CSV) relevant to the core concept of the method. Besides, our contribution is not about the model architecture, but the novel task, a theoretical framework for the solution, and its practical implementation.
> 10. **L15: The unit for "0.06 photons per pixel" is not specified (e.g., per frame or per second).**\
> We will clarify this to 0.06 photons per pixel per frame.
> 11. **L98: The meaning of s_i and i is unclear (intensity or pixel position?).**\
> We will clarify that i indicates the pixel position, and s_i indicates the Poisson rate at i.
> 12. **L112: The probability of zero photons hitting a pixel being e^{s_i} needs a detailed explanation.**\
> We explained this in A1 L445-463 in detail. We will clarify that in our revision.
> 13. **L197: The term "10-shot inference" is not explained.**\
> The 10-shot inference refers to the inference strategies in L243. This will be clarified in the revision.
> 14. **How does the method perform on extremely low photon count data, and what is the lower limit of photon count where the method remains effective?**\
> We simulated low photon count data with an average of 0.06 photons/pixel. Some data have even lower photon count (e.g., plasma data Fig. 2, S7, with photon count in the order of 0.001). Unclear what is considered extremely low or effective.
> 15. **How does the method perform on different types of scenes or motion patterns?**\
> Results cover a wide range of scene and motion patterns are provided in Fig. 1, 2, 5, S4-S14.

---

> > ### Comment · Reviewer_iaWM · 2024-08-13
> >
> > Thank you for addressing my concerns. While the reorganization of the response clarifies some aspects, it makes it difficult to follow my original concerns. Therefore, I've outlined my remaining comments below, following the original numbering for clarity.
> >
> > 1. [Comments] I am satisfied with the explanation provided for selecting the photon splitting variable p, which significantly impacts the method's applicability.
> >
> > 2. Many crucial implementation details about the dataset are absent.
> >
> >     [Comments] The response lacks crucial implementation details, particularly regarding dataset statistics. In lines 208-211, more specific information about the dataset size is needed, such as: "The proposed dataset contains XX real videos and XX synthetic videos, each with XX frames. Synthetic sequences were simulated from XX dataset videos using XX method." While Figures S5-S12 provide examples of real SPAD data and reconstruction, I am seeking a comprehensive statistical profile, not just illustrative examples. Without this information, the evaluation may be biased if it is based on a single video.
> >
> > 3. The comparison with existing quanta video reconstruction methods is limited to only QBP (proposed in 2020). There is a lack of quantitative comparisons with other existing methods.    Can you provide a more comprehensive comparison with other existing quanta video reconstruction methods beyond QBP, including quantitative results?
> >
> >     [Comments] The theoretical foundations of N2N, N2V, and GAP do not restrict their application to specific data dimensions. If I understand the proposed method correctly, it shares this flexibility. Therefore, the applicability to 3D data is not a unique contribution of the proposed method. Additionally, QBP can be adapted for 3D data by applying a densely sampled sliding window, which further emphasizes the need for a broader comparison.
> >
> > 4. The paper lacks a detailed analysis of the method's runtime and memory requirements compared to existing approaches. Can you provide a detailed analysis of the computational complexity and how it scales with dataset size? Could you elaborate on the runtime and memory requirements of your method compared to existing approaches?
> >
> >     [Comments] To clarify, I am seeking a fair quantitative comparison of inference computational complexity between different methods under the same input data size and hardware requirements, as commonly found in the literature. This typically involves evaluating the minimal memory and runtime required for inference. The current discussion only mentions the maximum GPU memory, total training time, and a rough estimate of minutes used without any constraints.
> >
> > 5. More extensive ablation studies exploring the impact of various components of the method (e.g., network architecture choices, hyperparameters) are needed to provide deeper insights into the method's performance. Can you provide more extensive ablation studies on the impact of various components of your method, such as network architecture choices and hyperparameters?
> >
> >     [Comments] As stated in line 57, one of the paper's main contributions is providing insights into network design. However, Table S1 only lists the selected hyperparameters, not the results of ablation studies. The only relevant ablation study is Table S7, which examines the effect of model size.
> >
> > 6. Several technical terms and abbreviations are not properly defined or explained:
> > - L1: "SPAD" is not defined on first use.
> > - L22: "QBP" is not defined on first use.
> >
> >     [Comments] I mean the first use in the main text, not the abstract. My suggestion is that the terminologies should be defined in the main text when they are used for the first time, even if they appear in the abstracts.
> >
> > 7. Some experimental details are missing.
> > - How does the method perform on extremely low photon count data, and what is the lower limit of photon count where the method remains effective?
> >
> >     [Comments] The statistics provided are about the simulation. The effectiveness means quantitative results demonstrating the method's performance under different illumination levels measured in lux.
> >
> > - How does the method perform on different types of scenes or motion patterns?
> >
> >     [Comments] Can you provide an related analysis including insights into potential limitations or guidelines for practical applications?

---

> > > ### Author Response · Authors · 2024-08-14
> > > **Response to the reviewer iaWM's new comments**
> > >
> > > We appreciate the reviewer’s last-minute response. We want to note another factual error in this comment:
> > >
> > > 6. **I mean the first use in the main text, not the abstract. My suggestion is that the terminologies should be defined in the main text when they are used for the first time, even if they appear in the abstracts.**\
> > > QBP is also defined on first use in the main text (L291).\
> > > We acknowledge that SPAD is not defined in L25. The initial definition is on the same page. We will fix it in the revision.
> > >
> > > Please find other responses below:
> > >
> > > 2. **The response lacks crucial implementation details ...**\
> > > We stated that each real SPAD dataset has 100-130k frames in L208. Each data set has a single video. We can provide other required information in the revision. We are not sure what the term “comprehensive statistical profile” means in this context.
> > >
> > > 3. **The theoretical foundations of N2N, N2V, and GAP ...**\
> > > We acknowledge that most convolutional neural networks can be applied to multidimensional data. In fact, we tested 3D N2N, N2V, and GAP with the same network architecture as shown in Table 1. However, directly applying these methods in 3D to 1-bit SPAD data produced suboptimal results. We also applied masked loss to 2D GAP (Table 1). It did not produce comparable results either. The combination of the 3D implementation, masked loss, and p-thinning-based sampling strategy, together with important architectural design choices such as group normalization, is the key to high-quality 1-bit quanta image reconstruction. The main contribution of this work is the combined concept and approach.
> > > While QBP can be adapted for 3D data by applying a densely sampled sliding window, the original manuscript indicates that speed (30 minutes/frame) is a major limitation as noted in L295. This is not a practically viable approach for reconstructing thousands of densely sampled sliding windows. Additionally, QBP works in a fundamentally different way than our method and we are not aware of how to make fair quantitative comparisons. It registers and bins adjacent frames, therefore the output is an accumulation of aligned images instead of a direct volume prediction. Despite that, we tested real SPAD data from QBP with our method and reported the result in Fig. 5. We are not aware of any method beyond QBP for a broader comparison.
> > >
> > > 4. **To clarify, I am seeking a fair quantitative comparison ...**\
> > > There are no differences in inference computational complexity for the 3D methods compared in Table 1. The only difference is the self-supervised training strategy. In L214, we mentioned “to ensure fair comparisons, we incorporated the same network architectures, hyperparameters, and training steps into individual baselines…”  Therefore, we believe we provided a fair quantitative comparison of inference computational complexity between different methods under the same input data size and hardware requirements.
> > > Also, this work highlights a new self-supervised concept for Bernoulli distributed data. Still, we provide a network design that is readily useful within the defined constraints. We acknowledge the network performance can be further improved using optimal architectures and hyperparameters, but it is beyond the scope of this work. The numerical information we provided in Table S1, L192, and L298 can be used to estimate the general runtime requirements in different situations by scaling the parameters.
> > >
> > > 5. **As stated in line 57, one of the paper's main contributions ...**\
> > > We conducted extensive ablation studies besides Table S7. Table S6 and Fig. 4d compare the initial filter size and indicate both large and small sizes can degrade the performance of the network. Fig. 4a and L261 compare the group normalization size, which is key for quality reconstruction. We also consider p as a highly relevant parameter in our method, and we conducted thorough ablation studies with different p ranges in Fig. 4b, c and Table S2-5, 8. We also included results with many different experiments in hp.CSV as noted in the Table S1 caption. All of the studies include numerical results. It is unclear which specific ablation studies the reviewer wants. \
> > >
> > > 6. **See above**
> > >
> > > 7. **The statistics provided are about the simulation ...**\
> > > We will clarify that the pixel values indicate the absolute illumination level (photon count). SPADs are very sensitive devices and work at the single photon level. Lux is a perceptual measurement of visible illuminance, which is not an appropriate measurement. For example, SPAD measures the spectrum through UV-NIR. Lux only accounts for visible light.\
> > > The requirements/assumptions for applying this method are discussed in L49-51. In extreme cases, as shown in Fig. S4, we applied the method to a random binary array with 1% of the pixels occupied with ones. The result is an expected output of a blank image. One limitation of the method is that if the pixels are not independent, the predictions can be erroneous.

---

> > > > ### Comment · Reviewer_iaWM · 2024-08-14
> > > >
> > > > Thank you for addressing my concerns and providing clarifications.
> > > >
> > > > The term 'comprehensive statistical profile' means exactly 'more specific information about the dataset size is needed, such as: "The proposed dataset contains XX real videos and XX synthetic videos, each with XX frames. Synthetic sequences were simulated from XX dataset videos using XX method." '
> > > >
> > > > Evaluating a method on a single video can lead to limited generalizability and biased results, which is unusual in the community. Therefore, I still have concerns on the evaluation conducted on a single video and the proposed datasets that each contains a single video. Robust evaluation necessitates using diverse datasets with multiple videos to demonstrate a method's effectiveness adequately. I would appreciate the authors' response to this specific concern. How do you plan to address this potential limitation and ensure the generalizability of your method?
> > > >
> > > > I am satisfied with your responses regarding the remaining points. I will raise my rating accordingly based on your responses and the promised incorporation of clarifications in the revision.

---

> ### Author Response · Authors · 2024-08-14
>
> We thank the reviewer for their comments.
>
> Regarding our used datasets, please note that we commit to publishing our datasets.
> We agree that the suggested clear and to-the-point summary of the used data can be helpful.
> We will add a sentence, in the form suggested by the reviewer, stating the number of videos and frames in the main paper:
>
> "To evaluate our method, we use a total of 7 real SPAD videos, each containing 100k-130k frames, and a synthetic video with simulated noise consisting of 3990 frames. Additionally, we use a real video with 100k frames published in [8]".
>
> We additionally will point to the description of the simulation in L199 “Simulated data” section.
> We will also add a table to the supplementary material describing the characteristics of each video including frame rate, number of frames, frame size (resolution), and subject content.
>
> However, we would like to stress that our manuscript includes experiments on a total of 9 (7 real + 1 synthetic + the video from [8]) videos, each showing vastly different content including low-signal, high-signal, high-contrast, high-ambient-light, moving camera, moving object, linear movement, random movement, combined movement, ultra-fast events, and stochastic events. The real data include dynamically moving objects, plasma balls, high-frequency bubble dynamics, histology images, and fluorescence microscopy data. The simulated data was carefully taken to cover a large range of image contrast and structures.
>
> We will include results on additional simulated data from other domains, including microscopy, in the supplementary material of the final version.

---

### Official Review · Reviewer_Z4Sm · 2024-07-14

**Soundness:** 3
**Presentation:** 3
**Contribution:** 3
**Rating:** 6
**Confidence:** 3

**Summary:**

The paper proposes to extend the Generative Accumulation of Photons that was proposed for Poisson noise to 1 bit Quanta image sensors.

**Strengths:**

The proposed method is novel, and mathematically interesting. The authors have put in a lot of effort to fit GAP for the problem of 1-bit QIS reconstruction.

**Weaknesses:**

The proposed method does not seem to be performing better than a supervised method. It is not clear what is the supervised method that was used in the comparisons.

It is not clear what the unique advantages of this method are compared to standard techniques like supervised learning or data simulation based training.

**Questions:**

Wouldnt randomly assigning each photon event to one of the two bins lead to loss of temporal information to some extent in both the input and target?
The supervised method seems to be performing better than the proposed method on static simulated scenes? Would it achieve similar fps as the proposed method on the video data too?

**Limitations:**

Yes. The authors have made the limitations of the proposed method clear.

---

> ### Author Rebuttal · Authors · 2024-08-05
>
> We appreciate the reviewer’s insightful comments and we want to answer the reviewer’s questions one by one:
>
> 1. **The proposed method does not seem to be performing better than a supervised method. It is not clear what is the supervised method that was used in the comparisons.**\
> We apologize for the confusion caused by the insufficiently detailed description of the supervised method. The purpose of the experiment was to characterize the theoretical upper limit of the network performance if clean ground truth was available. The supervised method was trained using pairs of clean ground truth data and simulated noisy data. The supervised method used the identical network architecture as our self-supervised method and was trained with cross-entropy loss. It ensured that the selected network architectures and hyperparameters could effectively represent the ground truth distribution. We will include that in the revised manuscript.
>
> 2. **It is not clear what the unique advantages of this method are compared to standard techniques like supervised learning or data simulation-based training.**\
> Supervised training has many limitations in practice, especially in scientific experiments (e.g., microscopy, ultra-high-speed imaging, etc.). Supervised training requires ground truth data that covers the distribution of the noisy measurements. Unlike natural image processing, very often we cannot acquire such a large amount of ground truth data, cannot acquire ground truth data at all, or cannot ensure the distribution of the measurements matches the distribution of the training data. The mismatch between the training data and the measurements can cause erroneous predictions. Our self-supervised method can restore the image using solely the information from the noisy dataset itself, making it appealing for scientific data processing.\
> Regarding data simulation-based training, if noise-free ground truth data is available, we can use simulated noise to produce the required training data. We believe the truncated Poisson noise model is sufficiently accurate for training the model. It may be possible to simulate the ground truth data given the knowledge of the underlying ground truth signal distribution. However, this knowledge is often unavailable in non-natural imaging applications. \
> *In summary, the self-supervised method does not require ground truth data or prior knowledge about the signal distribution, making it an effective solution for many scientific applications.*
>
> 3. **Wouldn't randomly assigning each photon event to one of the two bins lead to loss of temporal information to some extent in both the input and target?**\
> Each photon event has its own spatial-temporal index (pixel location and frame number). Assigning a photon event to one of the two bins (input and target) does not affect its frame number and pixel location, so it does not lead to loss of temporal information.
>
> 4. **The supervised method seems to be performing better than the proposed method on static simulated scenes? Would it achieve similar fps as the proposed method on the video data too?**\
> We apologize for the confusion. The simulated scenes presented in the work are also dynamic video data (see L204, Fig. S2). There are 3990 frames in the ground truth data with motion, recorded at 240 fps. See the response #1 above regarding the comparison of performance. Regarding the inference performance, the supervised method has the same network architecture as our method and outputs the same temporal resolution. Therefore the fps of inference is identical. We are happy to provide further clarification in the discussion phase of the rebuttal process.

---

### Official Review · Reviewer_abBE · 2024-07-15

**Soundness:** 4
**Presentation:** 4
**Contribution:** 3
**Rating:** 8
**Confidence:** 4

**Summary:**

This paper introduces a method to reconstruct/denoise high-resolution high-frame-rate videos captured by 1-bit quanta imaging sensors (e.g., SPAD arrays) without heavy spatio-temporal binning. The paper also captures and will release a new SPAD dataset.

The proposed method is loosely based on Generative Accumulation of Photons [9], which trains a CNN to reconstruct images in a self-supervised fashion by reconstructing the pdf of the photon arrivals. This paper makes several significant modifications to [9] so that it can work with quantum image sensors: First, in order to account for 1-bit sensors, the proposed method models the photon arrival pdf as a Bournoulli (rather than Poisson) distribution to account for the binary nature of the measurements. Second, in order to improve reconstruction accuracy it incorporates temporal information regularization.

The proposed method is tested on experimentally captured data and produces visually compelling results, including in the presence of non-rigid motion (e.g., guitar string).

**Strengths:**

A SPAD dataset would be extremely valuable to the research community.

Proposed method is novel and works well.

Paper is well written.

Denoising quanta image sensor data is an important problem with many scientific imaging applications

**Weaknesses:**

Proposed technique largely follows from GAP. Unclear if Poisson distribution assumption is invalid in the photon starved regime, i.e., would GAP work for dimmer scenes?

Figure/table captions could use more info. E.g., state whether Table 1 results are experimental or simulated.

Underperforms supervised methods.

**Questions:**

Does a Poisson signal model work in the low-flux regime (where it would be rare for more than one photon to arrive in a pixel)?

Can one bin a large # of frames in overlapping windows to ensure Poisson statistics while still recovering high temporal resolution?

How important is self-supervised training? Is the simulated noise model accurate enough that one could train an effective QIS denoiser using only simulated data?

**Limitations:**

Well discussed

---

> ### Author Rebuttal · Authors · 2024-08-05
>
> We appreciate the reviewer’s insightful comments and we want to answer the reviewer’s questions one by one. For conciseness, we grouped similar and relevant questions and comments.
>
> 1. **Is the Poisson distribution valid in the photon-starved regime (fewer than one photon per pixel on average)? How does GAP behave in this regime? How does the proposed method differ?**\
> It depends on the physics of the photon-counting process. If the sensor produces true Poisson photon counts, GAP should be applicable even for very low photon counts. Pixels can have more than one photon even if the Poisson rate is low. However, single photon detectors (like SPAD) are binary, leading to a truncated Poisson distribution / Bernoulli distribution (see L107). In this case, the Poisson distribution assumption becomes invalid at the pixel level leading to the artifacts shown in the main paper Page 7 Table 1 (Ours / No Mask, which is equivalent to a 3D version of GAP), where every pixel that has a photon in the input image leads to a darker result in the corresponding pixel in the output image (also see the figure in the rebuttal PDF). This is a consequence of correlations introduced by the splitting operation. Reducing the amount of light (and photons) will not remedy this problem, and the output images will still exhibit the same artifacts, albeit for fewer pixels, as there are fewer photons in the input. Addressing this issue is a major contribution of this manuscript. Our masking scheme can reliably avoid these artifacts even in low-light conditions.
>
> 2. **Can one bin a large # of frames in overlapping windows to ensure Poisson statistics while still recovering high temporal resolution?**\
> Binning a large number of binary frames will make the data behave more Poisson, but the temporal resolution cannot be recovered. There is a trade-off. As soon as we sum multiple frames, we can no longer distinguish which frame each photon is coming from. The temporal information is lost in the process. In our experiments, each frame has a 6 ns exposure time, but the frame rate is 100k fps (10 us/frame). There is a relatively long period between frames when the camera is not collecting light. If there is fast movement, the movement between frames will be integrated after binning, and binned frames will represent a much longer time scale, causing motion blur.
>
> 3. **Comparison to supervised methods and training using simulated data.**\
> We acknowledge that the supervised method is the upper limit of the network performance under ideal conditions. However, supervised training requires high-quality ground truth data that covers the distribution of the noisy measurements, leading to limited practical applicability. In many cases, we cannot acquire such a large amount of ground truth data, cannot acquire ground truth data at all, or cannot ensure the distribution of the measurements matches the distribution of the training data. The mismatch between the training data and the measurements can cause erroneous predictions.\
> If noise-free ground truth data is available, we can use simulated noise to produce the required training data. We believe the truncated Poisson noise model is sufficiently accurate to generate training data. In many applications, ground truth data is unavailable. In some cases, it may be possible to simulate the ground truth data, given the knowledge of the underlying ground truth signal distribution. This knowledge is often not available in non-natural imaging applications (e.g. scientific imaging).\
> *The ability to restore noisy data without ground truth data or prior knowledge about the signal distribution is a necessity for many scientific applications. Using a self-supervised method is important in this situation.*
>
> 4. **Figure/table captions could use more info. E.g., state whether Table 1 results are experimental or simulated.**\
> We will include more information in the Figure/Table captions in the revision. Table 1 results are from simulated data to have a ground truth for evaluation.

---

> > ### Comment · Reviewer_abBE · 2024-08-12
> >
> > Thank you for answering my questions.

---

### Author Rebuttal · Authors · 2024-08-06

We have addressed each reviewer's comments and questions in detail in reviewer-specific rebuttals underneath each review.

A new figure demonstrating the photon splitting process is presented in the attached PDF. The figure also more clearly indicates the artifact resolved by the masked loss. This figure is cited in abBE Response #1 and VmGM Response #4. We plan to add this figure to our revised manuscript.

---

### Decision · Program_Chairs · 2024-09-25

**Decision:**

Accept (poster)

**Comment:**

This paper received four positive leaning reviews --- one 5 (borderline accept), two 6s (weak accepts), and one 8 (strong accept).

There was broad appreciation for the novelty of the proposed method, quality of presentation, importance of the problem tackled (denoising and reconstruction from quanta sensor data), and the promise of a novel SPAD dataset

There were some concerns raised around the overall effectiveness of the proposed approach wrt supervised methods, and several other technical / exposition points which were reasonably well addressed in the rebuttal. There was considerable discussion post-rebuttal after which an accept consensus was reached. Therefore, an accept decision is recommended.

The authors are strongly urged to take into account the reviewers' concerns and include all the promised revisions when preparing the final camera-ready version of the paper.